# YAP1 nuclear efflux and transcriptional reprograming follow membrane diminution upon VSV-G-induced cell fusion

Daniel Feliciano [1✉], Carolyn M. Ott[1], Isabel Espinosa-Medina[1], Aubrey V. Weigel [1], Lorena Benedetti [1], Kristin M. Milano [2], Zhonghua Tang[2], Tzumin Lee [1], Harvey J. Kliman [2], Seth M. Guller[2] & Jennifer Lippincott-Schwartz [1✉]

Cells in many tissues, such as bone, muscle, and placenta, fuse into syncytia to acquire new functions and transcriptional programs. While it is known that fused cells are specialized, it is unclear whether cell-fusion itself contributes to programmatic-changes that generate the new cellular state. Here, we address this by employing a fusogen-mediated, cell-fusion system to create syncytia from undifferentiated cells. RNA-Seq analysis reveals VSV-G-induced cell fusion precedes transcriptional changes. To gain mechanistic insights, we measure the plasma membrane surface area after cell-fusion and observe it diminishes through increases in endocytosis. Consequently, glucose transporters internalize, and cytoplasmic glucose and ATP transiently decrease. This reduced energetic state activates AMPK, which inhibits YAP1, causing transcriptional-reprogramming and cell-cycle arrest. Impairing either endocytosis or AMPK activity prevents YAP1 inhibition and cell-cycle arrest after fusion. Together, these data demonstrate plasma membrane diminishment upon cell-fusion causes transient nutrient stress that may promote transcriptional-reprogramming independent from extrinsic cues.

[1] Janelia Research Campus, Howard Hughes Medical Institute, Ashburn, VA, USA. [2] Department of Obstetrics, Gynecology and Reproductive Sciences, Yale University School of Medicine, New Haven, CT, USA. ✉email: felicianod@janelia.hhmi.org; lippincottschwartzj@janelia.hhmi.org

Form and function are linked in both macroscopic and microscopic architecture. Rapid changes in cell shape and subcellular organization can initiate adaptive signaling pathways to shift cell bioenergetics and ensure survival. These changes also impact both cell specialization and cell proliferation[1–4]. Cell fusion, an essential process during regeneration and development, is a remarkable example of how cellular morphogenesis arising from the merging of two or more cells can create unique cell fates[5–7], but whether physical changes which occur during fusion have the ability to contribute to transcriptional changes that support a new cellular state remains unknown.

Systems undergoing cell fusion require special fusion proteins, or fusogens to assemble fusion pores and effectively merge plasma membranes (PM). Envelope viruses such as vesicular stomatitis virus (VSV), Zika virus, and SARS-CoV-2 utilize a similar process to gain access to eukaryotic cells. In many cases expression of fusogens in host cells after infection leads to viral-induced cell–cell fusion[8–11]. In mammals, cell fusion occurs in different tissues and organs including bone, skeletal muscle, immune cells, and placenta[6,7]. These systems have evolved specific fusogens that allow precise modulation of membrane fusion events. Interestingly, some of these fusogens originate from ancestral viral-fusing proteins that form part of the 8–10% of endogenous retroviruses genes encoded in the genome of humans and mice[12].

Prior to fusion, progenitor cells commit and differentiate into fusion-competent cells. This promotes fusogen surface expression and activation at the interface of adjacent PM[5,13–15]. Upon fusogen activation, PM and cytoplasmic content from fusing cells are combined to form the new syncytium. The resulting multi-nucleated syncytia can be comprised of hundreds to millions of cells, creating a unique cellular environment in which individual nuclei do not divide and acquire specialized functions[6,13–17]. This physical transformation suggests an underlying process could contribute to syncytial cell reprograming. This is in agreement with recent RNA-Seq studies in human placenta and mouse skeletal muscle cells revealing distinct transcriptional signatures between syncitia and mononucleated fusion-competent cells (e.g., cytotrophoblasts and myocytes, respectively)[13–17]. In vivo specification of each syncytium occurs in a complex environment containing extracellular signaling molecules. In several systems, some differentiation markers are present even when cell fusion is experimentally disrupted[18–21]. This has led to the view that syncytial specification is solely dependent on such extrinsic factors. However, it is possible that the unification of multiple cells by itself invokes cell-intrinsic pathways that contribute to their final transcriptional program[13–17].

Prior work has shown that cell fusion results in significant alterations in fundamental cell biological characteristics. These include changes in surface expression of membrane proteins, organelle intermixing, repositioning of nuclei, hormone secretion, and variations in metabolism including the regulation of the AMP-activated protein kinase (AMPK)[22–36]. It is unclear whether these and/or other cellular changes contribute to subsequent downstream transcriptional reprogramming after cell fusion. This knowledge gap stems from the challenge of isolating the intrinsic contribution of cell fusion from ever-present signaling molecules in organisms.

In this study we circumvent constraints by employing a VSV-G fusogen-mediated assay to induce cell fusion of culture cells in the absence of tissue-specific cues. Using this approach, we study the structural, subcellular, and transcriptional changes that occur to create a syncytium, and explore the molecular mechanisms involved. We characterize in detail the features of in vivo fusing systems replicated upon VSV-G-mediated cell fusion, including

fusion pore formation, PM mixing, and cytoplasmic mixing. In addition, a transcription regulatory factor, yes-associated protein 1 (YAP1), that promotes cell proliferation, vacates the nucleus upon cell fusion. The act of cell fusion induces changes in gene expression and prevents cell proliferation. In the absence of tissue-specific cues, transcriptional reprogramming after cell fusion results from remodeling the PM and a subsequent acute reduction in ATP levels that leads to the activation of AMPK and the downstream inhibition of YAP1.

## Results

**Cell fusion in a model system recapitulates physiological syncytial hallmarks.** To isolate cell fusion from cues existing within tissues, we employed an in vitro Vesicular Stomatitis Virus G protein (VSV-G) mediated fusion system to trigger cell fusion in culture cells[9,11,26,27,37]. In this system, VSV-G at the PM binds to the low density lipoprotein receptor (LDLR) on adjacent cells and upon washing with an isotonic, low pH buffer (Fusion Buffer) initiates PM fusion[38]. SUM-159 cells expressing endogenous LDLR were transfected with VSV-G and rapidly washed (5–10 s) with Fusion Buffer to induce fusion of two or more adjacent PM (Supplementary Fig. 1a). We estimate the transfection efficiency in our system to be near 60%. This was sufficient for widespread fusion events because VSV-G expressed on the surface is only required on one of the fusing cells[37]. Intracellular changes upon viral-mediated fusion systems have been described[9–11,26,27,37]. To define the timeline of physical cellular transformations in the cell lines and conditions we were using, we quantified changes in different fluorescent subcellular markers imaged for minutes, hours, and days and assessed how fusion re-shaped fundamental cellular features such as PM, cytoplasmic organization, transcription profile, and cell fate.

To assess how quickly and efficiently cell fusion occurred upon induction, we examined the speed of exchange of cytoplasmic proteins between fusing cells. To do this we monitored fluorescence intensity changes after fusing SUM-159 cells expressing only VSV-G (Receiver cells) or both VSV-G and a fluorescent cytoplasmic marker (Donor cells) (Fig. 1a, b, Supplementary Fig. 1b and Supplementary Movie 1). Within 30–60 s the fluorescence intensity of Receiver cells increased and Donor cell intensities started to decrease. This reflected the formation of fusion pores and the beginning of cytoplasmic mixing (indicated as Fusion in Fig. 1b). Full equilibration of the fluorescent cytoplasmic marker across the syncytium was achieved after 7–10 min (Fig. 1a, b and Supplementary Fig. 1b). The exchange of large subcellular organelles took longer (Supplementary Fig. 1c, d).

PM remodeling at the interface of fusing cells was examined using lattice light-sheet microscopy (Fig. 1c, d). Cells expressing VSV-G and the fluorescent PM marker, glycosylphosphatidylinositol anchored to mEmerald (mEmerald-GPI), were rapidly washed with Fusion Buffer to induce fusion of PM of two or more adjacent cells. Within seconds, the PM began to rearrange (Fig. 1c, d). Over several minutes the membrane boundary between adjacent cells disappeared as their PM coalesced (Fig. 1c lower panel and Supplementary Movie 3). During this process, the boundary between the two cells disappeared first in a small area near the coverslip and then propagated upward until only one cell outline was visible (all four experiments imaging this process show the same pattern of fusion). We also observed other morphological changes in the PM including dynamic membrane ruffling and the appearance of membrane projections emerging ~3–5 min after triggering cell fusion (Fig. 1c, d and Supplementary Movie 2 and Movie 3). Because actin dynamics are known to participate in pre- and post-fusion remodeling[39–43], we assessed

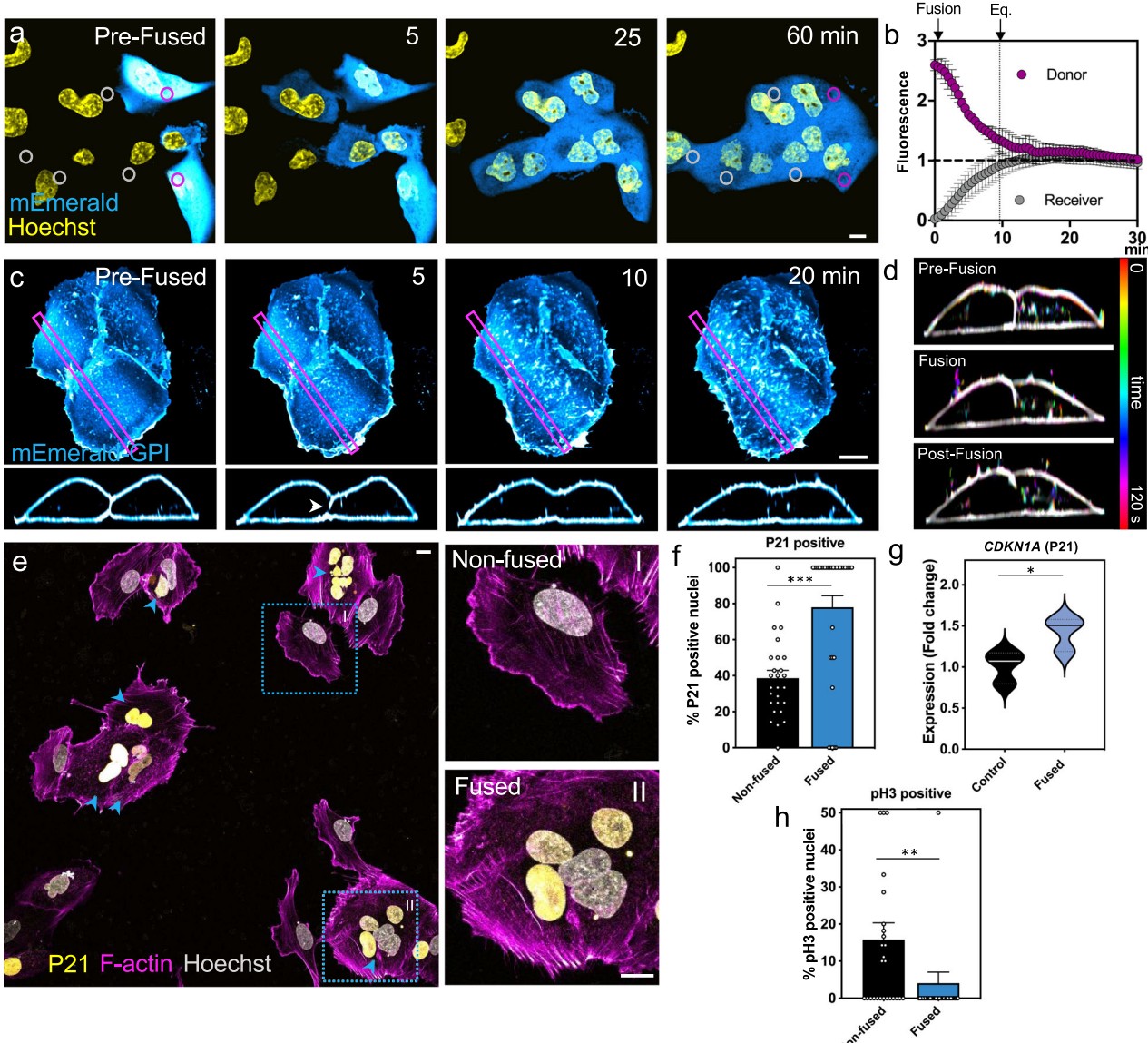

**Fig. 1 Changes in fundamental cellular features upon cell fusion lead to cell cycle exit. a** Cells expressing VSV-G alone (Receiver cells) or co-expressing VSV-G and a cytoplasmic marker (Donor cells, cytoplasmic mEmerald) were mixed and fused by a brief wash with Fusion Buffer and then were imaged by confocal microscopy. Cytoplasmic mixing was measured as fluorescence intensity changes in ROIs of Donor (magenta ROI) and Receiver (gray ROI) cells overtime upon fusion (see also Supplementary Movie 1). **b** Equilibration (Eq) of cytoplasmic mEmerald fluorescence intensity is marked with a vertical line. Error bars represent the standard error of the mean (SEM), $n = 4$ replicates for Receiver cells (gray dots) and $n = 2$ for Donor cells (magenta dots). **c, d** Cells expressing VSV-G and a plasma membrane marker (mEmerald-GPI) were induced to fuse. **c** Fusion pore formation (insets, white arrow head) and changes in the plasma membrane were assessed by lattice light-sheet microscopy. **d** Temporal color-coded images were generated by compressing 2 min from time points before (pre-fusion), during (Fusion), and after (post-fusion) cell fusion (see also Supplementary Movie 2). Images in **c** and **d** are representative of four experiments. **e** Representative confocal images of immunofluorescence staining of P21-positive nuclei in non-fused and fused cells (insets I and II, respectively) (arrow heads point to positive P21 nuclei in fused cells). **f** The percentage of P21-positive nuclei in non-fused (black bar) and fused cells (blue bar) quantified in cells imaged 24 h after washing with Fusion Buffer are graphed. Error bars represent the SEM, $n = 27$ and $n = 34$ cells examined over three independent experiments for non-fused and fused cells, respectively, ***$p < 0.0001$. **g** Violin plots depict the fold change in *CDKN1A* transcript (which codes for P21) expression quantified by qRT-PCR after 24 h in unwashed control (black) and fused (blue) cells. $n = 3$ independent experiments, *$p = 0.0412$. **h** 24 h after washing with Fusion Buffer, cells were stained for the protein pH3, a positive indicator of mitosis, and the percentage of pH3-positive nuclei in non-fused (black bar) and fused (blue bar) cells. Error bars represent the SEM, $n = 27$ and $n = 37$ cells examined over three independent experiments for non-fused and fused cells, respectively. **$p = 0.0279$. Statistical significance calculations were performed using a two-tailed unpaired Student's $t$ test. Scale bar size = 10 µm.

actin dynamics after fusion using a stable U2OS line expressing an F-actin marker (lifeact-EGFP) and observed that actin structures correlated with changes in PM remodeling (Supplementary Fig. 2 and Supplementary Movie 4).

Nuclei tracking analyses demonstrated that nuclei from fusing cells start to move toward the center of the newly formed syncytium within 10 min, and stably converge within 60 min (Supplementary Fig. 1e, f and Supplementary Movie 5).

Importantly, several of these subcellular changes, such as cytoplasmic mixing and nuclear clustering, have also been described in vivo after macrophage fusion and during the development of skeletal muscle and placenta[25,28,29,35,36].

While some nuclei can replicate in multinucleated cells such as the early *Drosophila* embryo, the nuclei of many mammalian syncytia arising from cell fusion lose their competence to enter the cell cycle[14,44,45]. This is consistent with the increase expression of the cell-cycle arrest regulator P21 (*CDKN1A)*, which is initially present in fusion-competent cells and is known to reach its highest levels after syncitia formation[46–50]. To test whether VSV-G-mediated cell fusion leads to increased P21 promoting cell-cycle arrest, we compared the levels of P21 in the nuclei of fused cells to the nuclei of non-fused cells in the same dish[46]. Non-fused cells include both non-transfected cells and cells expressing VSV-G that are remote from other cells. We find a twofold increase in P21-positive nuclei in fused cells 24 h after induction of cell fusion. This phenotype was specific to fused cells, since the percentage of P21-positive nuclei are unchanged in both untransfected cells washed with fusion buffer and unwashed cells expressing VSV-G (Fig. 1e, f and Supplementary Fig. 3). In addition, expression of P21 transcripts (*CDKN1A*), measured by qRT-PCR, was higher in fused cells than control cells that were transfected with VSV-G but left unwashed (Fig. 1g). In contrast, the nuclear levels of the positive mitotic marker pH3 are reduced by threefold in fused cells (Fig. 1h). We also assayed the frequency of syncytial division (segregation of nuclei into separate, smaller syncytia) after VSV-G-mediated fusion of SUM-159, U2OS, and 293T cells and observed that 88–95% of fused cells remain intact (Supplementary Fig. 4a, b). Cells in conflicting phases of mitosis are known to die upon fusion[45,51,52] and thus would lose adhesion and be largely excluded from our analysis. Furthermore, we did not observe changes in H2Ax (marker for DNA damage) in fused cells nuclei, therefore, it is unlikely that the change in proliferative state of fused cells is due to DNA damage (Supplementary Fig. 4c). These findings are consistent with an arrest in mitotic entry due to the increased expression of P21 (*CDKN1A*) in fused cells and suggest that the act of cell fusion, alone, can initiate transcriptional changes propelling a syncytium toward a differentiated-like state.

Altogether, the use of a VSV-G-mediated fusion system has allowed the detailed assessment of how cell fusion quickly remodels fundamental cellular features including cell shape (through PM and cytoskeleton dynamics), subcellular organization (through cytoplasmic and organelle intermixing), and the fate of the newly formed syncytium (directly restricting its capacity to enter the cell cycle). Furthermore, these results demonstrate that this model system is suitable to test whether the changes induced by fusion have the ability to trigger intrinsic pathways to modulate syncytial function because several characteristics observed in this system recapitulated those previously described in physiological syncytial systems[14,25,28,29,35].

**Cell fusion can induce transcriptional reprogramming toward a differentiated-like state.** Cell-cycle arrest is a representative characteristic that is coordinated with the reprogramming of gene expression during terminal differentiation[53]. To test whether cell fusion is sufficient to induce transcriptional reprogramming we performed RNA-Seq of unfused control cells and fused cells 6 h after washing with the fusion buffer (Fig. 2a). Dead cells and cellular debris were washed out prior to RNA isolation (see "Methods" section). Differential expression analyses using false discovery rate (FDR) < 0.05, revealed 3965 genes that differed between fused and control cells (2169 upregulated and 1796 downregulated in fused cells). Functional annotation clustering

using DAVID Bioinformatics Resources[54] of the gene ontology (GO) cluster of cellular components (GO:0005575) revealed that the majority of these genes are enriched in clusters of PM, vesicular, and cytoplasmic cell component genes (Fig. 2b), suggesting an overall structural remodeling that supports a new cellular state in fused cells.

To assess whether fused cells were changing their gene expression profile toward a differentiated-like state, we searched among all differentially expressed genes for either cell proliferation (GO: 0008283) or cell differentiation (GO: 0030154) related genes (Fig. 2c–f). Hierarchical clustering showed that a large cluster comprising 76% of the identified cell proliferation genes were downregulated (Fig. 2c, d). Conversely, 63% of identified cell differentiation-related genes were upregulated in fused cells (Fig. 2e, f). Global gene regulatory network analysis showed that upregulated cell differentiation-related genes are sub-divided into genes that promote both differentiation and development (Fig. 2g). Furthermore, in addition to downregulated genes promoting the cell-cycle, a subgroup of proliferation-related genes was upregulated. These genes are classified as negative regulators of cell proliferation, including P21 (*CDKN1A*) (Fig. 2h). This is consistent with our observation of syncytial cell-cycle arrest (Fig. 1e–h) and demonstrates that transcriptional reprogramming follows cell fusion.

**YAP1 is inhibited and redistributes from the nucleus into cytoplasm following cell fusion.** Transcriptional reprograming toward differentiation can be prevented by regulatory molecules that promote cell division. The Yes-associated-protein-1 (YAP1), when active, associates with various transcription factors and promotes cell proliferation[55,56]. Decreased YAP1 activity could permit reprograming toward a differentiated-like state by facilitating exit from the cellcycle. Interestingly, genes whose transcription is known to be regulated by YAP1 activity were among the differentially expressed genes identified by our RNA-Seq analyses (Fig. 3a). Transcription of most of these genes was downregulated by 15–45% in fused cells, suggesting YAP1 activity might be negatively regulated upon cell fusion (Fig. 3b).

The YAP1 activation state influences its subcellular localization; inactivation by the Hippo pathway and additional kinases such as AMPK prevents YAP1 nuclear import, resulting in redistribution of YAP1 to the cytoplasm and degradation[55,56]. Importantly, both extrinsic and intrinsic mechanisms can regulate YAP1 transcriptional co-activator activity[57,58]. To test whether cell fusion alters YAP1 activity, we looked at endogenous YAP1 localization in our VSV-G cell fusion system. Remarkably, we observed a shift in YAP1 distribution from the nucleus in non-fused cells to the cytoplasm in fused cells (Fig. 3c, top panel). Analyses of YAP1 localization at different time points revealed that relocation occurred within 1 h and was maintained at least up to 4 days after cell fusion (Supplementary Fig. 5 and Fig. 3c, d, top panel).

Given this significant response, we investigated whether YAP1 was also primarily localized in the cytoplasm in more physiological syncytia that do not require expression of an exogenous fusogen. For this, primary human trophoblasts purified from term placenta were cultured and allowed to fuse on their own for 2, 3, and 4 days[59]. Analyses of YAP1 localization revealed fused trophoblasts contain predominately cytoplasmic YAP1 localization similar to our cell fusion model system (Fig. 3c, d, bottom panel). These results are consistent with recent work that demonstrates that active, nuclear-localized YAP1 promotes maintenance of proliferating trophoblasts[60]. In addition, we assayed YAP1 localization in myoblast cells (C2C12 cells) that fuse in culture upon treatment with a differentiation media. YAP1

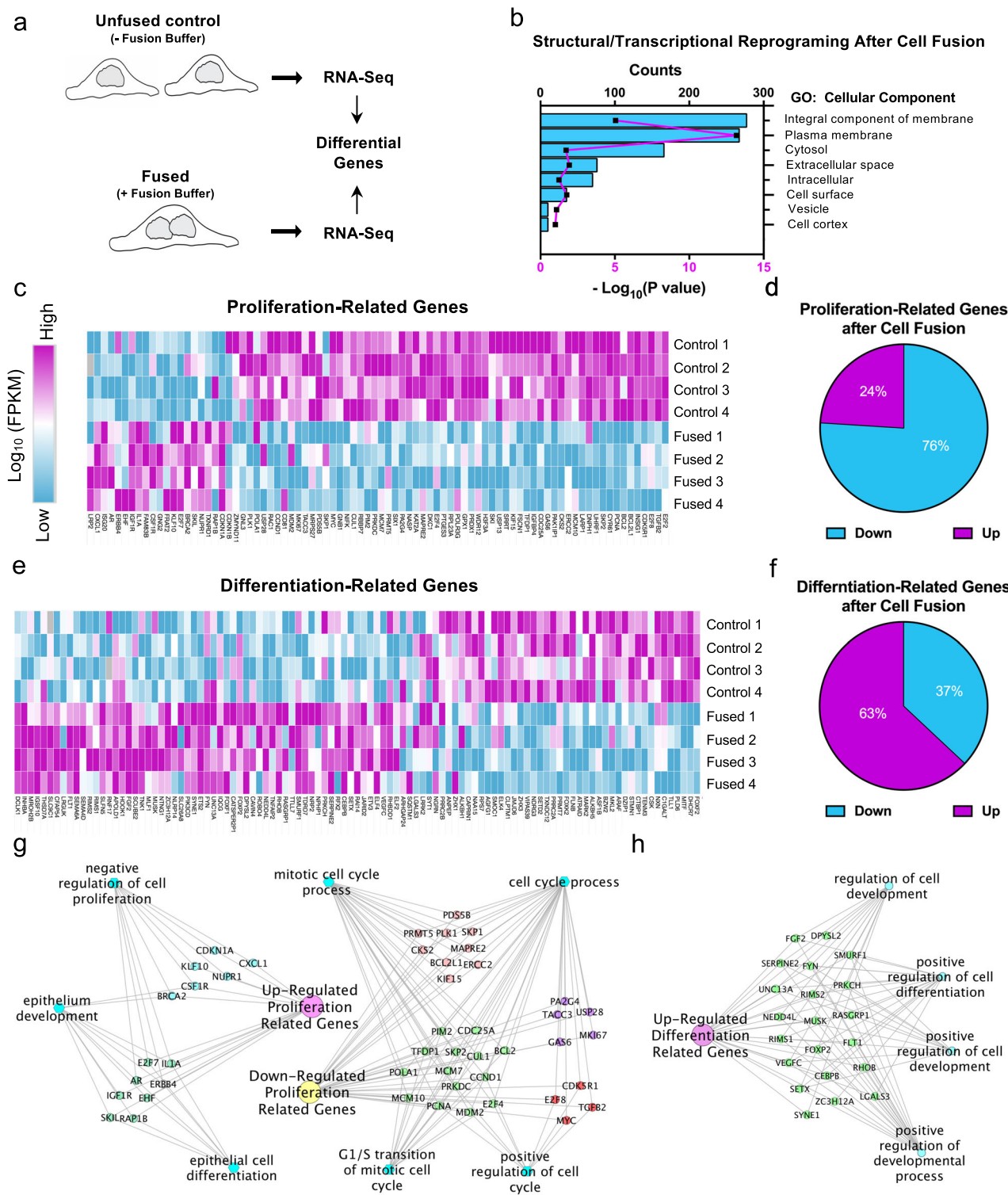

**Fig. 2 Cell fusion decreases the expression of proliferation-related genes while promoting expression of genes involved in differentiation and cell-cycle arrest. a** Graphical description of RNA-Seq work flow to compare cell transcriptional profiles of unwashed control cells to fused SUM-159 cells 6 h after induction of cell fusion. **b** Genes differentially expressed between fused and control cells were filtered by gene ontologies (GO). The differentially expressed genes were grouped into cellular component genes that significantly changed after cell fusion. **c–f** Differentially expressed genes were filtered to identify cell proliferation-related (GO 0008283) and cell differentiation-related (GO 0030154) transcripts. **c, e** Expression levels (Log10 FPKM) of four independent experiments are shown as heatmap visualizations of the percentage of proliferation (**d**) and differentiation (**f**) genes that were down or upregulated after cell fusion. **g, h** ToppCluster plots showing the functional network among differentiation and proliferation-related differentially expressed genes.

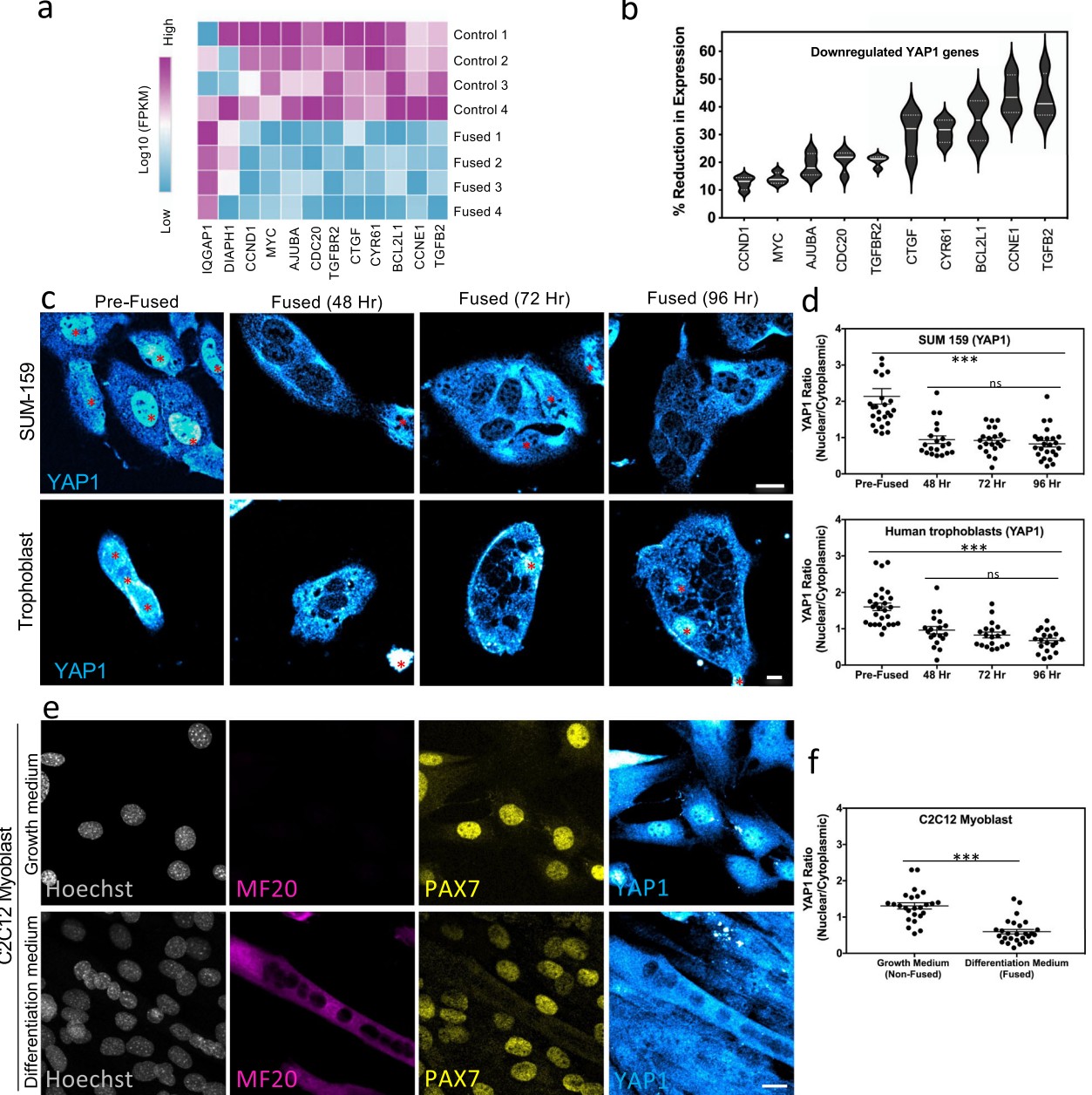

**Fig. 3 Cell fusion leads to YAP1 inhibition and cytoplasmic localization. a** Heatmap representing the expression levels (Log10 FPKM) of transcripts from genes known to be influenced by YAP1 activity in non-fused unwashed control and fused cells ($n = 4$ independent experiments). **b** Violin plots represent the average percent decrease in transcript levels from $n = 4$ independent cell fusion experiments. The analysis focuses on transcripts known to be influenced by YAP1 activity. Confocal images of YAP1 immunostained VSV-G transfected SUM-159 cells (**c**, top panel) and isolated primary human trophoblast cells that fuse in culture without induction (**c**, bottom panel) either before or after fusion were quantified (**d**) to measure changes in the localization of YAP1. Representative images are shown. Red * marks non-fused cells. (Top panel) Error bars represent the SEM, $n = 26$, $n = 20$, $n = 21$, and $n = 25$ cells examined over three independent experiments for pre-fused and cells 48, 72, and 96 h after fusion, respectively. ***$p < 0.0001$. (Bottom panel) Error bars represent the SEM, $n = 27$, $n = 19$, $n = 19$, and $n = 20$ cells examined over three independent experiments for primary human trophoblast prior to fusion or left to fuse in culture during 48, 72, and 96 h, respectively. ***$p < 0.0001$. **e** C2C12 myoblast fusion experiments were performed. Control cells incubated in growth media were fixed after 2 days. Cells induced to fused were incubated in differentiation media for 4 days and then were fixed. C2C12 cells were immunostained and imaged by confocal microscopy to determine YAP1 localization. Proliferative C2C12 were labeled using PAX7 antibodies while fused muscle cells were labeled using MF20 antibodies. **f** The nuclear/cytoplasmic ratio of YAP1 before and after cell fusion of C2C12 myoblast cells was calculated. Error bars represent the SEM, $n = 25$ and $n = 27$ cells examined over three independent experiments for cells in growth or differentiation media, respectively. ***$p < 0.0001$. ns not significant ($p > 0.05$). Statistical significance calculations were performed using a two-tailed unpaired Student's $t$ test. Scale bar size = 10 μm.

is mostly localized in the nucleus of PAX7-positive myoblasts and the nuclear to cytoplasmic ratio of YAP1 is significantly lower in the MF20-positive syncytia (Fig. 3e, f)[61]. We also examined YAP1 in tissue sections from mouse skeletal muscle in developing

mouse embryos (E10.5) and could see YAP1 present in the nuclei of PAX7-positive progenitor cells. In MF20-positive cells, overall YAP1 levels are reduced, consistent with YAP1 degradation which is known to follow YAP1 inhibition (Supplementary

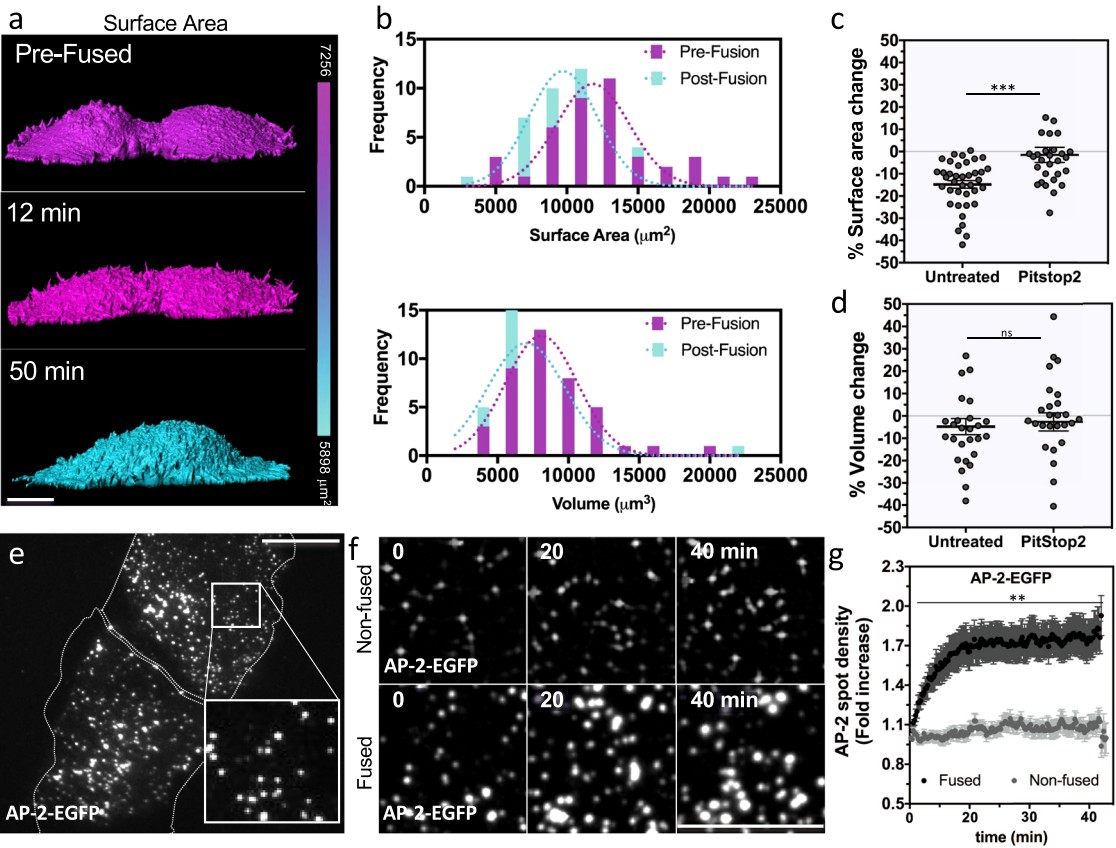

**Fig. 4 Remodeling upon cell fusion reduces the plasma membrane surface area by increasing endocytosis. a** SUM-159 cells expressing VSV-G and a PM marker (CAAX- EGFP) were imaged by confocal microscopy as cells fused (Z-stacks were used to generate three-dimensional models of cells). The surface of fusing cells is colored based on the surface area as measured by the IMARIS surface tool (see also Supplementary Movie 6). **b** The frequency distribution of surface area and volume of pairs of cells within 30 min of cell fusion are shown. The percentage change in plasma membrane surface area (**c**) and cell volume (**d**) 30 min after cell fusion with or without PitStop2 treatment to inhibit endocytosis are graphed. Error bars represent the SEM, $n = 38$ and $n = 29$ cells examined over three independent experiments for untreated and PitStop2 treated cells, respectively. ***$p = 0.0004$. **e, f** CRISPR-Cas9 gene edited SUM-159 cells expressing endogenous AP-2 –EGFP and transfected with VSV-G, were induce to fused, and the density of AP-2 –EGFP at the plasma membrane was measured overtime by TIRF microscopy (using 100 $\mu m^2$ cropped images, such as the inset in **e**). Representative images of AP-2 –EGFP puncta at the indicated time points in non-fusing (**f**, upper) and fusing cells (**f**, lower). **g** The fold increase in AP-2 –EGFP puncta density was measured every 16.5 s for about 1 h. The differences in AP2-EGFP density between non-fused and fused cells became statistically different 2 min after cell fusion began (see also Supplementary Movie 7). Error bars represent the SEM, $n = 13$ and $n = 41$ for non-fused and fused cells, respectively. From 2 to 44 min, *$p < 0.03$. ns not significant ($p > 0.05$). Statistical significance calculations were performed using a two-tailed unpaired Student's $t$ test. Scale bar size = 10 $\mu m$.

Fig. 6)[55,56]. Together, the trophoblast and myoblast investigations suggest YAP1 inactivation may be a conserved characteristic of multiple fused cell systems which contributes to transcriptional reprograming.

**Cell fusion promotes increased endocytosis altering plasma membrane surface area.** We next evaluated whether specific changes in fundamental cellular features occurring during cell fusion were responsible for YAP1 redistribution and transcriptional changes that follow VSV-G-induced fusion in SUM-159 cells. Changes in shape are known to be governed by alterations in both volume and surface area (SA)[62,63]. Importantly, recent work has shown that YAP1 activity and cellular localization is altered through variations in volume and/or SA[64–66]. Given our observation that fusing cells undergo a significant change in cell shape and active membrane dynamics, we wondered whether this could be related to changes in YAP1 localization upon cell fusion.

To measure SA during cell fusion, cells expressing a fluorescent PM marker were fused, imaged, and then used to create a 3D surface using the software Imaris. Immediately after cells start to fuse, membrane protrusions are observed. This was followed by a decrease in SA after several minutes (Fig. 4a and Supplementary Movie 6). To measure the extent that cell fusion alters SA and volume, we compared the SA and volume of cells before (Pre-Fusion) and 30 min after cell fusion (Post-Fusion) (Fig. 4b). We observed a 15% decrease in PM SA after cell fusion was induced, while volume only modestly changed (Fig. 4b–d). This suggests that upon fusion the newly formed syncytium activates adaptive responses that reduce its PM SA while keeping its volume relatively constant.

The role of endo-exocytic pathways in the regulation of PM SA has been well documented[67]. Therefore, our finding showing a decrease in PM SA suggests that upregulation of endocytosis might be mediating PM internalization. To test this hypothesis, the PM of cells were labeled with a lipid fluorescence dye (DiD) and the internalization of vesicles was monitored in fusing and non-fusing cells. We observed that upon cell fusion the number of internalized vesicles increases (Supplementary Fig. 7a). This supports the idea that upregulation of endocytosis during VSV-G-mediated cell fusion is responsible for PM SA reduction in these cells, which could potentially influence YAP1 cellular distribution and transcriptional regulation.

**Upregulation of clathrin-mediated endocytosis is necessary for surface area regulation and YAP1 inactivation during cell fusion.** Prior work has shown that clathrin-mediated endocytosis (CME) can regulate the SA of cells that are undergoing cell division[68,69]. To determine whether CME is also involved in the regulation of PM SA upon cell fusion, we assessed the frequency of clathrin-coated pits by TIRF microscopy. Analyses of PM clathrin-heavy-chain (CHC) and the alpha-subunit of the clathrin adapter, AP-2 showed that the levels of both on the PM increased after cell fusion, suggesting CME had increased (Supplementary Fig. 7b). To accurately measure the degree by which CME was augmented after the formation of fusion pores, we used a gene edited cell line expressing endogenous levels of fluorescently labeled AP-2[70] (Fig. 4e). Consistent with upregulation in CME, there was a 75% increase in AP-2 spot density 20 min after cell fusion, whereas non-fused cells showed no change in the levels of AP-2 spot density (Fig. 4f, g; see Supplementary Movie 7 for the determination of T = 0 min). Furthermore, pretreatment of fusing cells with the inhibitor of endocytosis, PitStop2, blocked the reduction in SA, suggesting that active CME is required for proper control of the PM SA after SUM-159 cells fuse (Fig. 4c).

Given the significant effect of endocytosis in reducing the PM SA early after cell fusion, we asked whether this was the upstream regulatory mechanism leading to YAP1 redistribution. To address this, we examined whether inhibiting CME blocks YAP1 cytoplasmic retention. Preincubation of fusing cells with either of two different inhibitors of endocytosis (PitStop2 or Dynasore) strongly decreased YAP1 redistribution into the cytoplasm (Fig. 5a, b). We confirmed these drugs do not affect normal VSV-G localization, consistent with the observed ability of inhibitor treated cells to fuse (Supplementary Fig. 8). Furthermore, expression of the CME-specific inhibitor AP180-C showed a higher impact on preventing YAP1 cytoplasmic redistribution, demonstrating the specific role of CME in YAP1 regulation after cells fuse (Fig. 5a, b)[71].

Since inhibition of CME strongly blocks YAP1 nuclear exclusion, we reasoned that upregulated endocytosis after cell fusion might alter transcription. Importantly, prior work has shown that down-regulation of YAP1 can lead to cell-cycle arrest and the upregulation of P21[72]. To test whether inhibition of endocytosis, which leads to YAP1 nuclear retention, blocks cell-cycle arrest we measured the levels of P21 in fused cells pretreated with PitStop2. We found that fused cells pretreated with PitStop2 have fewer P21-positive nuclei than untreated fused cells (Fig. 5c, d). Consistent with these findings, qRT-PCR analyses revealed that inhibition of endocytosis also reduced the expression levels of P21 (*CDKN1A*) in fused cells (Fig. 5e). This demonstrates that active endocytosis acts as a cell-intrinsic cue leading to YAP1 nuclear exclusion and the expression of cell-cycle arrest genes.

**Increased CME upon cell fusion results in transient glucose transporter depletion and acute energy stress.** Next, we investigated a possible intracellular mechanism downstream of CME that leads to YAP1 cytoplasmic redistribution. Active endocytosis not only regulates PM SA in cells, but is also essential for controlling the surface distribution of a wide-range of membrane proteins including surface receptors and transporters[73,74]. Recent studies have suggested that surface expression of glucose transporters and the levels of cytoplasmic glucose can control YAP1 localization and activity[75,76]. Therefore, one possibility is that increased CME upon cell fusion changes the distribution of glucose transporters leading to YAP1 nuclear exclusion. To test this possibility, we examined whether cell fusion triggers the internalization of glucose carriers.

The most widely expressed glucose transporter isoform, Glut1[77,78], localizes largely to the PM in SUM-159 cells. Upon fusion, we observe an accumulation of Glut1-positive internal structures ~5 min after cell fusion was triggered. These internal structures start decreasing after 15 min and equilibrate after 60 min as PM labeling of Glut1 returns to pre-fusion levels (Fig. 6a, top panel and insets, quantified in Supplementary Fig. 9a). This suggested Glut1 transporters are actively recycled back to the PM at these later times. Glut1 can be internalized by both CME and clathrin-independent endocytic (CIE) pathways[74]. To test whether inhibition of CME affects Glut1 internalization upon cell fusion, we analyzed the localization of Glut1 after fusion of cells previously transfected with AP180-C. Consistent with a specific role for CME during cell fusion, expression of AP180-C blocked the internalization of Glut1 in fused cells (Fig. 6b, lower panel and insets). Furthermore, an alternative localization analysis of the clathrin-dependent cargo transferrin receptor (TfR-GFP) using HEK 293T cells revealed TfR-GFP was also internalized 5 min after fusion (threefold) and, similar to glucose transporters, was partially recycled back to the PM after 60 min (Supplementary Fig. 9b). As expected, treatment with PitStop2 impaired TfR-GFP internalization (Supplementary Fig. 9b). Conversely, CD147 and CD98, two amino acid transporters known to be regulated by CIE, did not significantly internalize after cell fusion (Supplementary Fig. 10). These results demonstrate that upregulation of CME upon VSV-G-mediated cell fusion can directly and specifically influence the surface levels of Glut1 thus acutely modulating the PM landscape of the new syncytium.

Down-regulation of glucose transporters leads to reduced levels of cytoplasmic glucose and could lead to energy stress[79,80]. In addition, the many remodeling events that occur post-fusion require energy (for example, actin remodeling)[18]. To measure the levels of cytoplasmic glucose after cell fusion, we utilized a glucose biosensor, iGlucoSnFR.mRuby2 that increases or decreases its fluorescence intensity depending on whether it is in its bound or unbound state, respectively (Fig. 6c)[81]. We measured a rapid drop in cytoplasmic glucose that reached a minimum ~12 min after induction of cell fusion and then a gradual increase as cytoplasmic glucose levels recovered after 40 min (Fig. 6d). This result is consistent with our finding showing a fast internalization of glucose transporters within 5 min after cell fusion followed by recycling of the transporters back to the PM at later times (Fig. 6a). This glucose sensor does not directly measure changes in glucose flux through glucose transporters, but it shows that the decrease in glucose receptors at the surface correlates with reduced cytoplasmic glucose. Parallel to the results with the glucose biosensor, luciferase-based ATP measurements (detects the relative levels of ATP) detected a decrease in ATP levels 5 min after cell fusion followed by recovery after 60 min (Fig. 6e). This demonstrates that cell fusion induces an acute reduction in the energy state of the cell that cannot be quickly replenished because the glucose channels have been internalized by CME.

**AMPK acts as a downstream effector of cell fusion-induced structural changes to initiate gene reprogramming.** When cells are depleted of ATP, the AMPK is phosphorylated and activated[82]. AMPK is the master regulator of glucose metabolism and has the ability to sense the cytoplasmic AMP/ATP ratio. In addition, AMPK activity is important for cell differentiation-promoting processes including the regulation of YAP1[83]. To test whether the reduced energy state induced upon cell fusion leads to activation of AMPK, we determined the levels of phosphorylated AMPK (P-AMPK) by western blot analyses. We found a twofold increase in P-AMPK levels 5 min after cell fusion that

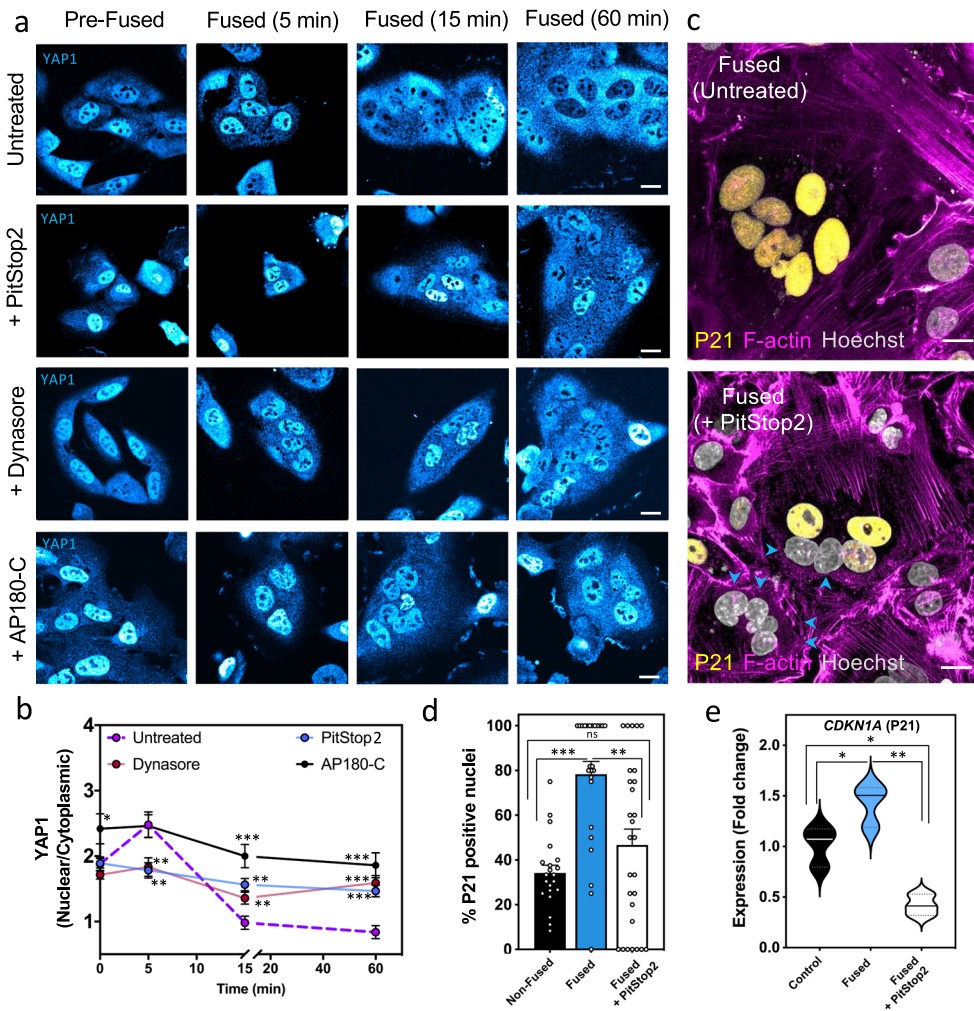

**Fig. 5 YAP1 inhibition and cell-cycle arrest after cell fusion depend on active clathrin-mediated endocytosis. a** SUM-159 cells, transfected with VSV-G, were incubated with inhibitors of endocytosis (PitStop2 or Dynasore) or transfected with a dominant negative form of the clathrin adapter AP-180 (AP180-C), a specific inhibitor of CME. Cells were then induced to fuse, fixed at indicated time points, immunostained with anti-YAP1 antibody and imaged by confocal microscopy to determine the YAP1 localization. **b** The ratio of nuclear to cytoplasmic YAP1 was measured at each timepoint after fusion in control (untreated) or endocytosis inhibited cells washed with fusion buffer. Error bars represent the SEM of cells examined over three independent experiments. For untreated cells (magenta line): $n = 44$, $n = 19$, $n = 21$, and $n = 23$ cells at 0, 5, 15, and 60 min, respectively. For Dynasore treated cells (red line): $n = 38$ (ns), $n = 25$ (**$p = 0.0092$), $n = 32$ (**$p = 0.0095$), and $n = 31$ cells (***$p < 0.0001$) at 0, 5, 15, and 60 min, respectively. For PitStop2 treated cells (blue line): $n = 101$ (ns), $n = 36$ (**$p = 0.0021$), $n = 44$ (**$p = 0.0003$), and $n = 50$ cells (***$p < 0.0001$) at 0, 5, 15, and 60 min, respectively. For cells transfected with AP180-C (black line): $n = 24$ (*$p = 0.019$), $n = 24$ (ns), $n = 25$ (***$p < 0.0001$), and $n = 28$ (***$p < 0.0001$) at 0, 5, 15, and 60 min, respectively. $p$ values represent the differences between the control and each endocytic inhibitor at the indicated time point. **c** Confocal images of P21-positive nuclei in untreated and PitStop2 treated cells fixed 24 h after fusion. Arrow heads points at nuclei within fused cells negative for P21. **d** Quantification of the percentage of P21-positive nuclei in both fused (blue bar) and non-fused (black bar) cells in the untreated samples and the fused cells in the PitStop2 treated samples (white bar) are graphed. Error bars represent the SEM, $n = 21$, $n = 25$, and $n = 28$ cells examined over three independent experiments for non-fused, fused, and PitStop2 treated fused cells, respectively. ***$p < 0.0001$, and **$p = 0.0012$. **e** Violin plots depict the fold change in CDKN1A (P21) expression measured by qRT-PCR and compared in unwashed control cells (black), fused cells (blue), and fused cells treated with PitStop2 (white). $n = 3$ independent experiments. *$p = 0.0412$, *$p = 0.0109$, and **$p = 0.0007$. ns not significant ($p > 0.05$). Statistical significance calculations were performed using a two-tailed unpaired Student's $t$ test. Scale bar size = 10 μm.

was maintained for up to 60 min and decreased thereafter (Fig. 7a, b). This is consistent with our measurements of Glut1 internalization and subsequent return to the PM (Fig. 6a). Since the reduced energy state induced by cell fusion is correlated with the transiently increased internalization of glucose channels and reduction of cytoplasmic glucose, we tested a casual relationship by examining whether inhibition of endocytosis blocks the activation of AMPK. For this we pretreated fusing cells with PitStop2 and measured the levels of P-AMPK. Similar to pretreatment with the AMPK inhibitor, compound C, inhibition of endocytosis by PitStop2 blocked AMPK phosphorylation (Fig. 7b). This

demonstrates that activation of AMPK upon cell fusion lies downstream of PM remodeling events.

Prior work has shown that AMPK can negatively regulate YAP1 when cells experience a low energy environment[84–86]. We reasoned that AMPK activation could link the structural and energetic changes occurring during cell fusion to the downstream inhibition of YAP1 in fused cells. To test whether YAP1 nuclear exclusion upon cell fusion requires active AMPK, we treated fusing cells with the AMPK inhibitor, compound C, for 3 h and measured endogenous YAP1 distribution. Remarkably, inhibition of AMPK completely blocked YAP1 re-localization to the

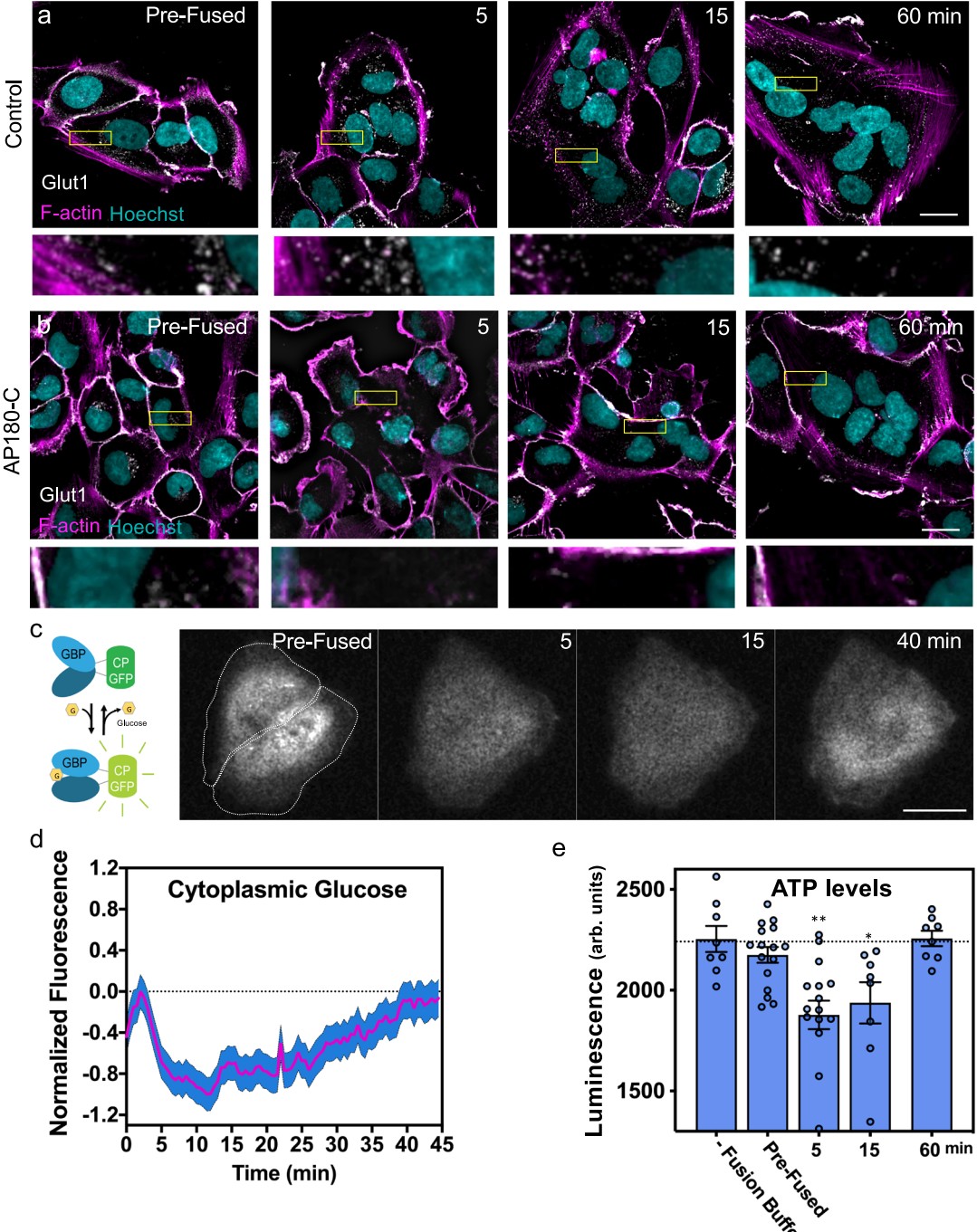

**Fig. 6 Cell fusion leads to a reduced energy state through transient, CME-dependent glucose channel internalization.** SUM-159 cells expressing VSV-G (**a**) or VSV-G and AP180-C (**b**) were induced to fused and then fixed at indicated time points. Anti-Glut1 antibodies were used to assess the subcellular localization of Glut1 by confocal microscopy. Insets depict a region in the cytoplasm at each time point. Images in **a** and **b** are representative of three independent experiments. **c** The glucose biosensor, iGlucoSnFR.mRuby2 (depicted in left panel, GBP glucose binding protein) fluoresces when glucose is bound. The intensity of the cytoplasmic biosensor fluorescence was monitored as cells expressing VSV-G were induced to fuse. Images are representative of nine independent experiments. **d** Relative cytoplasmic glucose levels were measured overtime in fusing cells. To calculate the relative changes, the fluorescence intensity in fusing cells was normalized to the intensity of control non-fusing cells. The blue area represent the SEM overtime, $n = 19$ cells. **e** VSV-G transfected SUM-159 cells were fixed before fusion, or at the indicated time after fusion was induced. Relative cytoplasmic ATP levels were measured using a luciferase-based assay (arb. units; arbitrary units). Error bars represent the SEM, $n = 8$, $n = 16$, $n = 16$, $n = 8$, and $n = 8$ replicates examined over two independent experiments for unwashed pre-fused, and fused cells after 5, 15, and 60 min, respectively. $p$ values represent the differences between the pre-fused cells and each time point (5 and 15 min). **$p = 0.0009$, and *$p = 0.0153$. Statistical significance calculations were performed using a two-tailed unpaired Student's $t$ test. Scale bar size $= 10\,\mu m$.

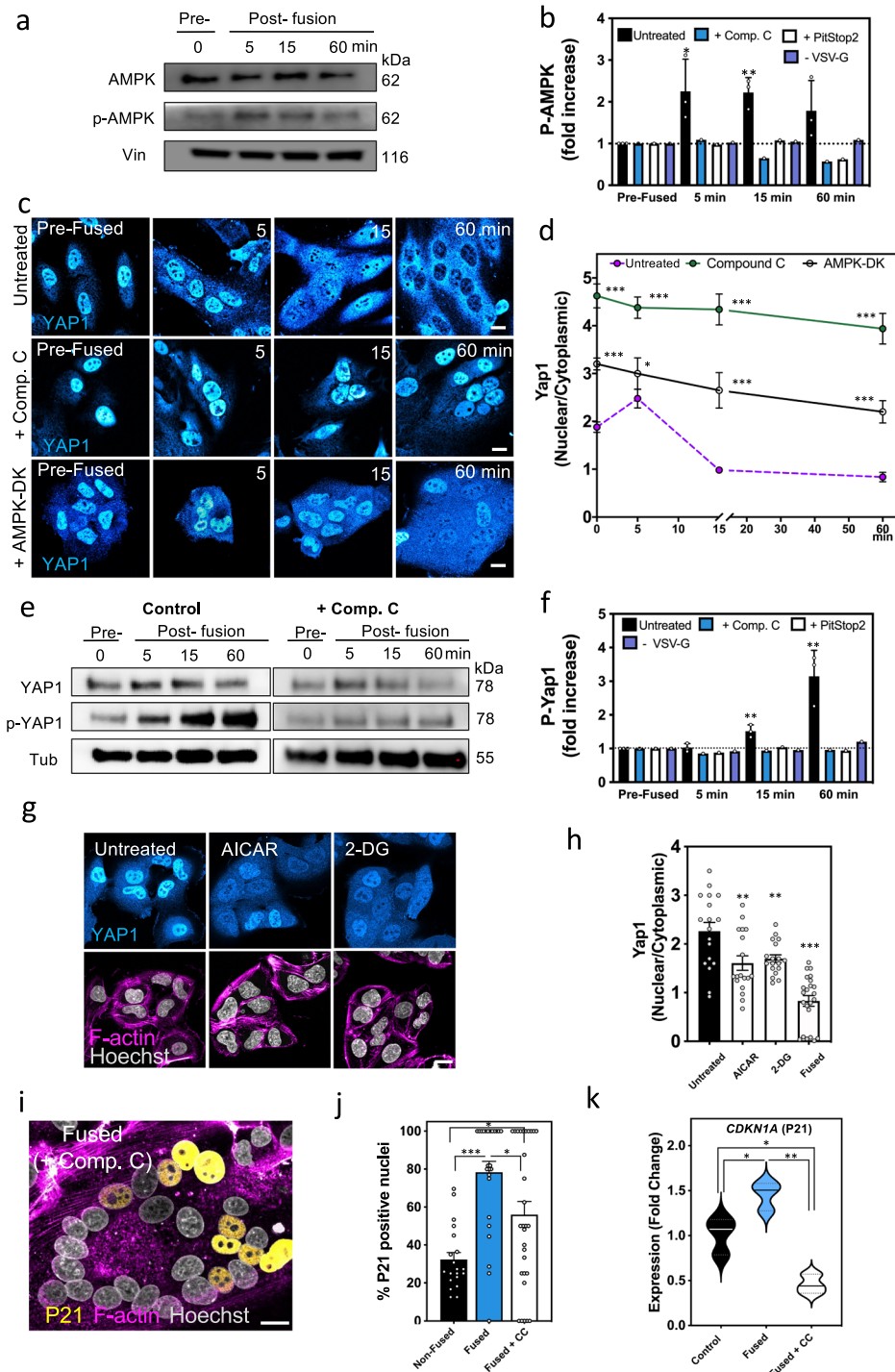

cytoplasm and led to a predominant nuclear localization of YAP1 in both fused and non-fused cells (Fig. 7c, d). We did not observe any effect of compound C on cell fusion or VSV-G localization (Fig. 7c and Supplementary Fig. 11a). YAP1 nuclear retention was also observed in fusing cells transfected with the dominant negative form of the AMPK α2 subunit (dead kinase: DK) confirming that active AMPK is necessary for YAP1 inhibition and redistribution to the cytoplasm upon cell fusion (Fig. 7c, d and Supplementary Fig. 11b). We also assayed a specific modification of YAP1—phosphorylation at S127—which is the target of the large tumor suppressor kinase and is known to be promoted by active AMPK[69]. Biochemical analysis showed that

fusing cells treated with compound C or endocytic inhibitors had lower levels of inactive S127 P-YAP1 compared with non-treated cells (Fig. 7e, f). We confirmed that inactivation of YAP1 and long-term cytoplasmic retention depends on both active AMPK and the act of cell fusion, since activation of AMPK alone (using AICAR or 2-DG) in unfused cells did not recapitulate the strong YAP1 cytoplasmic phenotype observed in cells that undergo fusion (Fig. 7g, h).

To test whether inhibition of AMPK blocks cell-cycle arrest, we measured levels of P21 in fused cells pretreated with compound C. Similar to PitStop2 treatment, we found that fused cells treated with compound C had fewer P21-positive nuclei compared to

**Fig. 7 AMPK is activated upon cell fusion and AMPK inhibition impairs both YAP1 cytoplasmic localization and cell-cycle arrest. a** Cell lysates were made from VSV-G- transfected SUM-159 cells before (pre-fusion; 0 min) or 5, 15, and 60 min after washing with fusion buffer. The levels of total (~62 kDa) and phosphorylated (~62 kDa) AMPK were determined by western blot. Vinculin (Vin, ~116 kDa) was used as a loading control. Western blot images are representative of three independent experiments. **b** The fold change of p-AMPK in lysates at the indicated times after fusion was calculated relative to the pre-fusion cells. All cells were transfected with VSV-G and were then left untreated (black bar), or treated with AMPK or endocytosis inhibitors (compound C, blue bar or PitStop2, white bar). In addition, p-AMPK was quantified in untreated cells that lacked VSV-G (untransfected cells washed with Fusion Buffer but unable to fuse, purple bar). Error bars represent the SEM, $n = 3$, $n = 3$, $n = 3$, and $n = 3$ independent lysates samples from pre-fused cells and cells isolated 5 (*$p = 0.049$), 15 (**$p = 0.004$), and 60 min after fusion, respectively. **c, d** SUM-159 cells transfected with VSV-G, incubated with or without the AMPK inhibitor compound C or co-transfected with the dominant negative form of the α2 subunit of AMPK (AMPK-DK), were induced to fuse and then fixed at indicated time points and immunostained with an anti-YAP1 antibody (**c**). The YAP1 nuclear to cytoplasmic ratios were graphed (**d**). Error bars represent the SEM of cells examined over three independent experiments. For untreated cells (magenta line): $n = 44$, $n = 19$, $n = 21$, and $n = 23$ cells at 0, 5, 15, and 60 min, respectively. For compound C treated cells (green line): $n = 51$ (***$p < 0.0001$), $n = 25$ (***$p < 0.0001$), $n = 32$ (***$p < 0.0001$), and $n = 24$ cells (***$p < 0.0001$) at 0, 5, 15, and 60 min, respectively. For cells transfected with AMPK-DK (black line): $n = 38$ (***$p < 0.0001$), $n = 26$ (ns), $n = 23$ (***$p = 0.0001$), and $n = 26$ cells (***$p < 0.0001$) at 0, 5, 15, and 60 min, respectively. **e** Cell lysates from VSV-G transfected cells, either untreated or incubated with compound C, were prepared at the indicated times after fusion. Western blot images are representative of three independent experiments for untreated samples. Total YAP1 (~78 kDa) and YAP1 phosphorylation (S127, phosphorylated by LATS, ~78 kDa) were assayed by western blot. (Tubulin (Tub), 55 kDa) was used as a loading control). **f** The fold changes of p-YAP1 were calculated in untransfected (-VSV-G, purple bar) cells, and in VSV-G-transfected cells that were untreated (black bar) or incubated with either compound C (blue bar) or PitSop2 (white bar). Error bars represent the SEM, $n = 3$, $n = 3$, $n = 3$, and $n = 3$ independent lysates samples from pre-fused cells and cells isolated 5, 15 (**$p = 0.0084$), and 60 (**$p = 0.0086$) min after fusion, respectively. **g** Unfused cells treated with AICAR (6 mM) or 2-DG (50 mM) for 1 h were fixed, immunostained with anti-YAP1 antibody and imaged by confocal microscopy to determine YAP1 localization. **h** The ratio of nuclear to cytoplasmic YAP1 measured in fused cells and control (untreated, black bar) or treated cells is graphed. Error bars represent the SEM of cells examined over three independent experiments, $n = 18$, $n = 18$, $n = 18$, and $n = 23$ cells for untreated, AICAR treated (**$p = 0.008$), 2-DG treated (**$p = 0.0067$), and fused cells (***$p < 0.0001$), respectively. **i, j** Nuclear P21 was visualized in compound C treated cells fixed and immunostained 24 h after fusion. **j** The percentage of P21-positive nuclei in compound C treated cells (white bar) is compared to the untreated non-fused (black bar) and fused cells (blue bar). Error bars represent the SEM, $n = 20$, $n = 25$, and $n = 30$ cells examined over three independent experiments for non-fused, fused, and compound C treated fused cells, respectively. ***$p < 0.0001$, *$p = 0.0119$, and *$p = 0.0189$. **k** Violin plots depict the fold change in *CDKN1A* (P21) expression levels in compound C treated (white) and untreated cells measured by qRT-PCR. $n = 3$ independent experiments (control, black and fused, blue). *$p = 0.0412$, *$p = 0.0138$, and **$p = 0.0008$. ns not significant ($p > 0.05$). Statistical significance calculations were performed using a two-tailed unpaired Student's *t* test. Scale bar size = 10 μm.

untreated fused cells (Fig. 7i, j). Furthermore, qRT-PCR analysis of compound C treated cells indicated that AMPK inhibition decreased the expression of P21 (*CDKN1A*) (Fig. 7k). Altogether, these results suggest VSV-G-mediated fusion of SUM-159 cells leads to an acute activation of AMPK promoting YAP1 nuclear exclusion supporting a new cellular state. This process is triggered by remodeling of the PM upon cell fusion independent from tissue-specific cues.

## Discussion

Cell fusion in vivo involves fast rearrangements of cell structural features and long-term transcriptional reprogramming[6,14–17,22–25,28–35]. How the initial structural changes that accompany cell fusion are integrated to alter cell fate determination is still unknown. This gap in understanding stems from the challenges in dissecting the intrinsic contribution of cell fusion from extrinsic differentiation signals within tissues. Here, we employed a viral fusogen (i.e., VSV-G) to induce fusion of cultured cells in the absence of tissue-specific differentiation cues. We identified a mechanism regulating key transitions that cause fused cells to stop proliferating and divert toward a differentiated-like state. We observed that cell fusion shifted the SA-to-volume ratio and triggered CME to remove excess membrane. As a result, glucose transporters (Glut1) were temporarily internalized leading to reduced cytoplasmic glucose and ATP levels. The transient drop in cytoplasmic ATP, activated AMPK, which promoted the phosphorylation of YAP1 at S127, and led to YAP1 inhibition and retention in the cytoplasm. Consequently, the transcription of several genes involved in cell proliferation dropped, and genes involved in cell-cycle arrest and differentiation were expressed. Furthermore, we showed that disruption of either CME or AMPK activation prevents subsequent parts of the pathway from operating. Importantly, the key feature—YAP1 inhibition and its exclusion from the nucleus—is recapitulated in fused human primary trophoblasts, C2C12 myoblast cells and developing muscle tissue.

These results demonstrate that physical and structural changes upon cell fusion are sufficient to trigger an intrinsic response that changes how transcription is regulated.

Functional differentiation of syncytia during skeletal muscle and placenta development requires an initial differentiation event where progenitors transition into fusion-competent cells that later fuse to fully differentiate[61,87]. In vivo studies have demonstrated that non-autonomous signaling is sufficient for partial differentiation progression in some systems[18–21]. However, it is also true that cell fusion is essential for the generation and maintenance of healthy skeletal muscle and placenta[20,88–91]. Previous studies have not examined the possibility that cell fusion in the absence of developmentally relevant signals could contribute to differentiation. Here, we have shown cell fusion in a model system can induce transcriptional reprograming in the absence of tissue-specific cues. Although myocytes and cytotrophoblasts express P21 and are thought to exit the cell cycle before they fuse[47–50], our observations suggest that additional programmatic changes in transcription, such as the increase expression of P21, could be facilitated by the physical processes that establish the final syncytia. This is in line with the idea that functional differentiation of syncytial systems is promoted by the act of cell fusion. Importantly, monitoring differentiation of syncitia in vivo in normal and fusion defective systems has been limited by staining for specific markers[18–20,92]. Transcriptomic approaches similar to the ones used here could be employed to identify defects in transcriptional programs in fusion defective systems and define the direct role of non-autonomous signals for syncitia differentiation[13–17].

CME is the primary endocytic pathway regulating PM SA during fundamental processes such as cell division[68,69]. Multiple links between endocytosis and different stages of cell fusion have been identified[93,94]. Examples of this include *C. elegans* cell fusion where the PM levels of the fusogen EFF-1 are controlled by endocytosis[94].

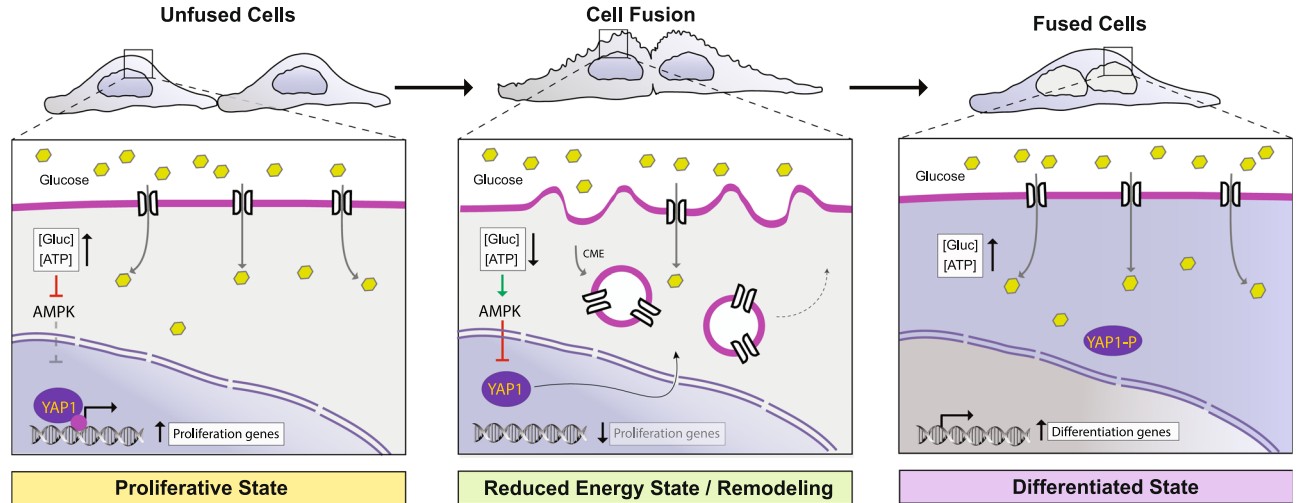

**Fig. 8 Structural remodeling upon cell fusion leads to endocytosis and AMPK-dependent YAP1 inhibition, which drives cell-cycle arrest and promotes a differentiated-like state.** In proliferating cells, nuclear YAP1 promotes expression of genes to support the proliferative state. The acute structural remodeling upon cell fusion, including endocytosis of glucose transporters, causes transient changes in cell energetics (decreased cytoplasmic glucose and ATP) and AMPK activation that lead to persistent retention of YAP1 in the cytoplasm. As a result, fused cells exit the cell cycle and transcripts that promote cell differentiation are generated.

How does cell fusion induce increased CME? Prior work has demonstrated that the level of tension of the PM regulates the equilibrium between exocytosis and endocytosis[67,95–97]. Specifically, exocytosis is stimulated by high membrane tension to add more membrane, while endocytosis is stimulated in response to low membrane tension[95]. We speculate, therefore, that excess PM at the interface where two cells have fused is sensed as low membrane tension, and that this cell intrinsic signal triggers increased CME during cell fusion.

AMPK has been shown to influence endocytosis and exocytosis of membrane proteins including ion channels and nutrient transporters[98–103]. Activation of AMPK is very sensitive to increases in intracellular AMP/ATP levels, which promote AMPK phosphorylation (in the range of 1.5 to 4-fold increase) within minutes[102,104]. Under energy stress, AMPK can increase the surface expression of Glut1 and Glut3 by promoting exocytosis[102,105]. We show that Glut1 is internalized 5 min after the onset of cell fusion and is eventually recycled back to the PM. Similarly, we measured a decrease in ATP levels 5 min after cell fusion, which led to rapid activation of AMPK. This was followed by ATP and glucose returning to initial levels. It is possible that activated AMPK promotes the recycling of glucose transporters to the PM to restore glucose levels. This is consistent with our measurements of cytoplasmic glucose using a glucose biosensor showing that recovery of cytoplasmic glucose occurs minutes after AMPK activation and coincides with recycling of glucose transporters. While temporally regulated AMPK activation appears to be necessary, AMPK activation in isolation is not sufficient: activation of AMPK by AICAR and 2-DG does not reproduce the dramatic and persistent depletion of YAP1 from the nucleus observed in fused cells. This is consistent with studies of unfused cells where AICAR and 2-DG treatment induced partial cytoplasmic localization of YAP1 that was reversed after drug removal[84–86]. Along with AMPK activity, additional pathways triggered by cell fusion may contribute to "flip a switch" that sets in motion the adaptive response we have described here.

YAP1 can directly control cell renewal by localizing to the nucleus, where it promotes proliferative transcriptional programs[56,57,106]. Prior work has shown that down-regulation of YAP1 can induce cell-cycle arrest and upregulation of P21[72]. In non-fusing cells, cytoplasmic localization of YAP1 has been strongly associated with differentiation of multiple cell types including keratinocytes, adipocytes, and neurons[107]. However, in fusing cells, little is known about YAP1 distribution and its role during reprograming. Prior work in vitro suggested that myoblast differentiation leads to YAP1 cytoplasmic localization and degradation[108]. Here we demonstrated that YAP1 in vivo is downregulated upon murine muscle fusion and differentiation (Supplementary Fig. 6). In addition, we demonstrated that in both primary trophoblasts and C2C12 myoblasts YAP1 localization in fused and non-fused cells paralleled the YAP1 nuclear-to-cytoplasmic shift observed in the VSV-G-mediated fusion system (Fig. 3c–f). Consistent with the results reported in this study, YAP1 transcritional activity in the nucleus of trophoblasts promotes maintenance of proliferation[60]. YAP1 inhibition and cytoplasmic redistribution is emerging as an important step and potentially a hallmark of syncytia differentiation triggered by cell fusion.

In summary, our data provide new insights into how intrinsic cellular changes arising from the fusion of two or more cells alter transcription. We describe a structural-to-transcriptional signaling pathway mediated by an endocytosis-AMPK-YAP1 axis that links membrane remodeling and cellular bioenergetics to transcriptional reprogramming in response to cell fusion (Fig. 8). In this pathway, AMPK plays a central role in sensing the transient reduction in cytoplasmic glucose and ATP caused by modifications in the PM landscape through increased CME. It then converts these changes into an adaptive response that inhibits YAP1 activity, inducing cell-cycle arrest and supporting the expression of differentiation-related genes. Hence, disabling either CME or AMPK by genetic or pharmacologic approaches hinders a transcriptional program toward a differentiated-like state during cell fusion. The broad transcriptional changes revealed by RNA-Seq, which include many transcripts not regulated by YAP1, suggest this pathway is likely integrated with additional unexplored cellular signaling routes to achieve the cell state transition. Future work will be needed to reveal whether this structural-to-transcriptional signaling pathway functions in tissues where it would synergize with additional environmental cues to accomplish functionally differentiated syncytia.

# Methods

**Cell lines**. The human mesenchymal triple-negative breast cancer-stem cell lines, SUM-159 and SUM-159-AP2-EGFP, were obtained as a gift from Tomas Kirchhausen. The C2C12 myoblast, U2OS, and HEK 293T cell lines were obtained directly from ATCC (CRL-1772, HTB96, and CRL11268, respectively). Stable U2OS expressing Lifeact-EGFP were generated in the lab using antibiotic (G418) resistance selection (cells were pooled and kept in antibiotic until used for experiments). Cells were grown in a 37 °C, 5% $CO_2$ tissue culture incubator on tissue culture treated dishes. All cell lines, with the exception of C2C12 myoblasts, were cultured in DMEM + 10% FBS, L-glutamine, and antibiotics. C2C12 were culture in growth medium (DMEM + 20% FBS, L-glutamine, and antibiotics). To induce C2C12 myoblast fusion, cells were grown in growth media until they reached 80% confluency, then were washed twice with 1x PBS and cultured with differentiation medium for 4 days (DMEM + 2% horse serum, L-glutamine, and antibiotics, Note: differentiation media was replaced every day). Cell lines were passaged with 0.25% Trypsin EDTA.

**Plasmids**. The pMD2.G VSV-G (#12259), H2B-mCherry (#20972), H2B-Halo Tag (#91564), TfR-EGFP(#54278), mEmerald (#53976), and p-AMPK alpha2 K45R (#15992) plasmids were obtained from Addgene. The AP180-C plasmid was a gift from Julie G. Donaldson. The plasmids for CAAX-EGFP (Farn-119), mTagBFP2-C1 (CV-261), Mito-EGFP (Clon-109) were obtained from the Michael Davison collection. iGlucoSnFR.mRuby2 was a gift from Loren Looger at Janelia Research Campus.

**Transfection**. All cell lines were transfected with VSV-G and the corresponding expression vectors using Lipofectamine 3000 (Thermo Fisher, Cat. # L3000015) following the manufacturer's protocol. Transfection efficiency for SUM-159 cells was about 60%.

**VSV-G mediated cell fusion**. Cells transfected with VSV-G and cultured at 37 °C, 5% $CO_2$ were rapidly washed (5–10 s) with an isotonic low pH buffer (125 mM NaCl, 5 mM KCl, 20 mM NaOAc, pH 5.5–6.0, Fusion Buffer 37 °C) to induce fusion of the PM of two or more adjacent cells. After fusion was induced cells were rapidly returned to regular medium and imaged at 37 °C, 5% $CO_2$ in a Zeiss LSM880 confocal microscope (for detailed protocol see Feliciano et al. [27]). For immunofluorescence analysis, cell fusion was induced sequentially at different times, and then all samples were fixed simultaneously in order to obtain the different time points of the fusion process. For immunoblot analysis, cells were fused and rapidly returned to the incubator (37 °C, 5% $CO_2$). After the indicated incubation time (0, 5, 15, 60 min) cells were placed at 4 °C to slowdown intracellular processes and scraped off to be used for cell lysates. Cellular debris in the Hoechst channel were removed from images by generating and then subtracting an inverted nuclei mask.

**Human trophoblasts**. Placentas from uncomplicated term pregnancies were collected within 30 min following elective cesarean section without labor at New Haven Hospital. Infection was excluded on the basis of standard clinical criteria (absence of fever, uterine tenderness, maternal/fetal tachycardia, foul vaginal discharge). Written informed consent was obtained from all participants before enrollment. Gestational age was established based on menstrual date confirmed by sonographic examination before 20 week gestation. Tissues were digested by trypsin/DNase I treatment and primary human trophoblasts were isolated. Primary trophoblasts were cultured in coverslip chamber slides and allowed to fuse for 48, 72, and 96 h. After the corresponding time point, pre-fused cytotrophoblasts and fused syncytiotrophoblasts were fixed with 4% paraformaldehyde (PFA, EMS) and used for subsequent immunofluorescence experiments (for detailed protocol see Kliman et al. [59] and Tang et al. [109]). Approval was granted by Yale University School of Medicine Human Investigation Committee.

**Histology of mouse embryos**. Ten-week-old pregnant CD-1 female mice were obtained from Charles River and the copulatory plug was labeled as day 0.5 dpc. At 10.5 dpc the mice were sacrificed by cervical dislocation and embryos were fixed for 3 h in 4% PFA (EMS). Immunofluorescence on sections was performed as follows: embryos were embedded in a 15% sucrose and 7.5% gelatin solution, frozen at −80 °C and sectioned (25 μm) using a Leica Cryostat. Primary antibodies were applied overnight in a PBS-Triton-FCS solution (PBS, 0.1% Triton X-100, 20% FCS). The slides were washed 3x for 10 min in PBS-Triton X-100 (0.1% Triton X-100), and secondary staining was performed in PBS-Triton-FCS containing Hoechst 33342 for 2 h at room temperature. Slides were mounted with Fluoromount (Sigma, Cat. # F4680) and imaged using a Zeiss LSM880 confocal microscope. The following primary antibodies were used: anti-Pax7, Mouse IgG1, (DSHB, ID:AB528428, 1:100), anti-MF20, Mouse IgG2b (DSHB, ID:AB2147781, 1:100), and Anti-Yap1 (Cell Signaling 14074S, 1:50). The following secondary antibodies were used: anti-mouse IgG2b Cy3 (Jackson Immunoresearch Laboratories 115-165-207, 1:500), anti-mouse IgG2b A647 (Jackson Immunoresearch Laboratories 115-605-207, 1:500), and anti-rabbit IgG A488 (Jackson Immunoresearch Laboratories 111-545-144, 1:500). Animals were singly housed, and were provided with food and water ad libitum. They were kept with a 12 h dark/12 h

light cycle in a temperature controlled (20–22 °C, humidity: 30–70%) and sound attenuated room. All animal experiments were conducted according to the National Institutes of Health guidelines for animal research. Procedures and protocols (17-152) on mice were approved by the Institutional Animal Care and Use Committee at Janelia Research Campus, Howard Hughes Medical Institute.

**Immunofluorescence**. SUM-159 cells plated on coverslip chambers (Thermo Fisher, Cat. # 155379) were fused at their corresponding time points, fixed with 4% PFA (EMS), and then permeabilized and blocked with blocking solution (0.5% Triton X-100, 10% BSA in PBS) for 1 h. Primary antibodies were diluted in blocking solution and incubated overnight at 4 °C. After 3 washes (10 min) with PBS, cells were incubated with Alexa Fluor-conjugated secondary antibodies. The following primary antibodies were used: Anti-Yap1 (Cell Signaling 14074S, 1:250), Anti-p21 (Cell Signaling, Cat. # 2947S, 1:200), Anti-pH3 (anti-Phospho-Histone H3 (Ser10); Cell Signaling, Cat. # 3377, 1:200), Anti-clathrin heavy chain (Abcam, Cat. # ab21679, 1:200), Anti-AP-2 (Abcam, Cat. # ab189995, 1:200), Anti-Glut1 (Abcam, Cat. # ab40084, 1:100), Anti-CD98 (BioLegend, Cat. # 315602, 1:200) and Anti-CD147 (BioLegend, Cat. # 306202, 1:200). For imaging and quantification, at least a total of 15 fields of view were randomly chosen by Hoechst 33342 nuclear staining (Thermo Fisher, Cat. # 62249) and imaged by Zeiss LSM880 confocal or NIKON TIRF microscope. At least three different samples were quantified per treatment type at each respective time point.

**RNA sequencing**. Prior to the isolation of RNA, dead cells and cellular debris were washed out with fresh medium. To Isolate the RNA, fused (washed with fusion buffer) or control (unwashed control) SUM-159 cells were lysed with TRIzol reagent (Thermo Fisher, Cat. # 10296010). A second $CHCl_3$ extraction was performed to increase RNA purity. Concentration and purity was determined by Nanodrop (Thermo Fisher). RNA-Seq libraries were made from 5 ng RNA per sample, using Ovation RNA-Seq v2 (NuGEN) to make cDNA and Ovation Rapid DR Multiplex System (NuGEN) to make libraries according to the manufacturer's protocol. ERCC Mix 1 spike-in controls (Thermo Fisher) were added at $1e^{-5}$ final dilution. Libraries were pooled for sequencing on a NextSeq 550 instrument (Illumina) using 75 bp reads in paired-end mode. Sequencing reads were trimmed to remove TruSeq adapters using Cutadapt (https://doi.org/10.14806/ej.17.1.200), then were aligned to the human genome (Hg38) using STAR (https://doi.org/10.1093/bioinformatics/bts635). Transcript BAMs were generated by STAR and gene expression estimates were made using RSEM (https://doi.org/10.1186/1471-2105-12-323). Differential expression analysis was performed using EBseq (https://doi.org/10.18129/B9.bioc.EBSeq) with FDR = 0.05. Gene enrichment analyses were performed using DAVID Bioinformatics Resources (URL: https://david.ncifcrf.gov, https://doi.org/10.1038/nprot.2008.211 and https://doi.org/10.1093/nar/gkn923). We used ToppCluster (URL: https://toppcluster.cchmc.org/, https://doi.org/10.1093/nar/gkq418) and Cytoscape (URL: https://cytoscape.org, https://doi.org/10.1101/gr.1239303) to construct the subcategory network. Heatmaps of gene expression were generated using the Morpheus software (URL: https://software.broadinstitute.org/morpheus/).

**RNA isolation and quantitative real-time PCR**. Relative gene expression was determined using TaqMan RNA-to-Ct 1-step kit (Thermo Fisher) with TaqMan gene expression assays for *CDKN1A* (Thermo Fisher) (Supplementary Table 1). The RNA from fused (washed with Fusion Buffer) or control (unwashed control) SUM-159 cells, that were treated or untreated with the endocytic inhibitor PitStop2 (Abcam, Cat. # ab120687) or the AMPK inhibitor compound C (Sigma, Cat. # P5499-5MG), was isolated using TRIzol (Thermo Fisher, Cat. # 10296010). qRT-PCR reactions were initiated with 100 ng of RNA for each sample following manufacturers protocol. Data were acquired with a Roche480 light cycler. Samples were run on triplicate plates and their Ct values averaged. Relative quantitation was performed using Delta-Delta Ct method. Analysis was performed using phosphoglycerol kinase (*PGK*) or glyceraldehyde 3-phosphate dehydrogenase (*GAPDH*) as a reference gene.

**Immunoblot analysis**. SUM-159 cells that had been scraped off plates as described above were lysed with lysis-buffer (50 mM Tris-Cl, pH 7.5, 150 mM NaCl, 0.1% SDS, 1 mM DTT, and 1 mM EDTA) containing protease and phosphatase inhibitors (Sigma) for 30 min at 4 °C. After lysing the cells, samples were centrifuged at $16,000 × g$ in a microcentrifuge and the supernatants were recovered for subsequent steps. Lysates were separated by SDS-PAGE on gels and transferred to PVDF. Blots were incubated with anti-YAP1 (Cell Signaling 14074S, 1:1000), anti-phospho-YAP1 (Cell Signaling (S127) 13008, 1:1000), anti-AMPK (Cell Signaling, Cat. # 2532s, 1:1000), anti-phospho-AMPK (Cell Signaling, Cat. # 2531s, 1:1000), anti-Vinculin (Sigma, Cat. # V9131, 1:5000), and anti-Tubulin (Sigma, Cat. # T9026, 1:5000). Secondary antibodies conjugated to HRP were used (Thermo Fisher, Cat # MA5-15367, 1:1500, and Sigma, Cat # 12348, 1:2000).

**ATP measurement**. To determine the relative levels of cytoplasmic ATP at different time points during the fusion process, total cellular ATP was assayed using a luciferase-based ATP determination kit following the manufacturer's protocol (Thermo Fisher, Cat # A22066).

**Surface area and volume measurements**. To determine the SA and volume, SUM-159 cells transfected with VSV-G and a PM marker (CAAX-EGFP) were culture at low confluency and only pairs of cells adjacent to each other were imaged before and 30 min after cell fusion. PitStop2 (Abcam, Cat. # ab120687) treated and untreated cells were imaged to acquire Z-stacks using a Zeiss LSM880 confocal microscope. Determination of SA and volumes was achieved using the surface tool in the Imaris software (Bitplane).

**Glucose biosensor**. Images of cells expressing VSV-G and the iGlucoSnFR. mRuby2 glucose biosensor were acquired on a Zeiss LSM880 confocal microscope immediately after cell fusion was induced. While the green channel was used to monitor cytoplasmic glucose levels during the fusion process, the red channel served to confirm that cells were stable during the experiment in the $x$–$y$ focal planes.

**TIRF microscopy**. Live cell and immunofluorescence imaging of clathrin and AP-2 at endocytic sites was performed by TIRF microscopy using a NIKON Eclipse Ti Microscope System equipped with an environmental chamber (temperature controlled at 37 °C and $CO_2$ at 5%), Apo TIRF ×100 objective (NA 1.49), high-speed EM charge-coupled device camera (iXon DU897 from Andor), and NIS-Elements Ar Microscope Imaging Software.

**Bromophenol blue (BPB) quenching assisted microscopy**. HEK 293T cells transfected with both VSV-G and the transferrin receptor (TfR-EGFP) were pre-cultured with or without PitStop2 (Abcam, Cat. # ab120687). Total fluorescence and Bromophenol blue (BPB)-quenched images were taken at different time points upon fusion (0, 5, 15, 60 min) and the Total/Internal fluorescence ratios were calculated. For the quenching step, BPB (Sigma, Cat. # B8026) was dissolved in phenol red-free DMEM containing 25 mM HEPES buffer (Thermo Fisher, Cat. # 15630080) and applied at a final concentration of 2 mM to ensure instant and effective quench of EGFP fluorescence.

**Airyscan microscopy**. To visualize changes in F-actin during cell fusion, U2OS cells stably expressing Lifeact-EGFP were transfected with VSV-G, and washed with Fusion Buffer. Imaging was performed using a Zeiss LSM880 with Airyscan microscope with a plan-apochromatic ×63 oil objective (NA = 1.4). Images were processed with Airyscan processing in ZEN black software (Zeiss). Bleach correction (histogram matching) was applied using image J.

**Lattice light-sheet microscopy**. To monitor PM dynamics upon cell fusion, cells transfected with VSV-G and a PM marker (mEmerald-GPI) were washed with Fusion Buffer before imaging on a custom-built square lattice light-sheet microscope (LLSM). Imaging was performed using 488 nm excitation at an exposure time of 0.018 s (250 slices = 4.8 s), and a multiband pass emission filter (NF03-405/488/532/635E, Semrock). The annulus was set for an outer NA of 0.5, and an inner NA of 0.42. Data were acquired by serial scanning of the entire fusing cells through the light sheet. To ensure minimum photobleaching/phototoxicity, a pause was added between acquisitions for a final 3D imaging rate of 30 s per volume. All acquired data were deconvolved by using a Richardson–Lucy algorithm adapted to run on a graphics-processing unit, using an experimentally measured PSF.

**Statistics and reproducibility**. No statistical methods were used to predetermine the sample sizes. Some experiments were randomized and the investigators were blinded to allocation during experiments and outcome assessment. At least 15 images were taken for each experiment involving immunofluorescence. The sample sizes were considered sufficient given that large differences, with a $p$ value lower than 0.01, between the two experimental conditions were usually detected. All of the quantitative data shown represent the mean ± SEM, except when otherwise stated in the legend. Bar plots have been overlaid with dot plots showing all of the individual measured data points. No strongly scattering data points were excluded; all quantitative evaluation data points were taken into account and averaged to fully represent biological and technical variabilities.

Statistical analyses were done using the GraphPad Prism 8 software. Statistical significance calculations comparing two conditions were performed using a two-tailed unpaired Student's $t$ test. The experiments were repeated at least three times and were reproducible.

**Reporting summary**. Further information on research design is available in the Nature Research Reporting Summary linked to this article.

## Data availability

Data supporting the findings is available in the article, Supplementary Information, and the microscopy data are available upon request. All RNA-seq data generated as a part of this study have been deposited in the National Center for Biotechnology Information Gene Expression Omnibus (GEO) and are accessible through the GEO Series accession number GSE168125. Source data are provided with this paper.

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

## Acknowledgements

We thank Andrew Lemire, Jingqun Ma, and Kshama Aswath for help with RNA-seq data generation; Kevin McGowan for help with the qRT-PCR data generation; Tomas Kirchhausen for providing the SUM-159 and SUM-159-AP2-EGFP cells; Jacob Keller and Loren Looger for sharing the iGlucoSnFR.mRuby2 plasmid; Julie G. Donaldson for the AP180-C plasmid; Ulrike Heberlein and Erik Snapp for helpful discussion and comments on the manuscript. We thank the JLS lab for helpful suggestions. The work was supported by the Howard Hughes Medical Institute.

## Author contributions

D.F. and J.L.S. designed experiments and supervised the work. D.F. performed and analyzed in vitro experiments. D.F. and I.E.M., with support from T.L., performed and analyzed immunofluorescence experiments on mouse embryos. D.F. and I.E.M. analyzed RNA-Seq data. D.F. and I.E.M. created model in Fig. 8 with feedback from other authors. D.F. and L.B. performed and analyzed experiments measuring the percentage of syncytia division. D.F., K.M.M., Z.T., H.J.K., and S.M.G. performed human trophoblast purification and fusion experiments. D.F. and A.V.W. performed LLS microscopy experiments. D.F., C.M.O., and J.L.S. wrote the manuscript.

## Competing interests

The authors declare no competing interests.
