## [Peer Review File · Nature Communications]

REVIEWER COMMENTS

Reviewer #1 (Remarks to the Author):

The work by Feliciano and co-authors addresses the question of which cellular processes are regulated as a consequence of cell-cell fusion. Using cancer cells in culture, the authors show that viral VSV G-induced cell fusion and the increased plasma membrane surface triggers a homeostatic mechanism of clathrin mediated endocytosis; a low energy cellular state which activates AMPK pathway; the re-localization of YAP1 to the cytoplasm affecting the expression of diverse genes; and a cell-cycle arrest. The study appears to be carefully conducted and the results are interesting. However, there are major related issues that the authors have to address before I can recommend it for publication.

Major comments:

1. A major concern is the generalization of the results obtained in very specific experimental conditions: cell-cell fusion induced by the viral fusogen VSV G and low pH employing mainly the breast cancer cell line, SUM-159. The authors should perform more experiments to demonstrate that the processes described in this study are conserved in other systems (e.g. different cell types, including stem and primary cells lines) and cell-fusion events induced by different ways (e.g. fusogens that do not require acidification, chemical/electrical fusion). Alternatively, the authors could re-write the manuscript to avoid making general conclusions. Even though this occurs throughout the Ms. the generalization is reflected clearly in the title, which should have some additions: "Cell fusion (induced by VSV G) triggers membrane diminution causing transcriptional reprogramming (in cancer cells in vitro)".
2. The studies of VSV G as a fusogen are classic and researchers are still actively working to understand the mechanisms and consequences of VSV G-mediated fusion. The authors have not cited this extensive work in their introduction and also not in the discussion (for example White et al., 1981; Abou-Hamdan et al., 2018)
3. The present work is based on a gain-of-function system to show a link between a cell-cell fusion event and the up-regulation of genes related to differentiation (down-regulation of genes related to proliferation). However, some loss-of-function studies reported that knockout myoblasts for Myomaker or Myomerger fail to fuse but still differentiate, suggesting that fusion does not directly regulate differentiation (reviewed in Sampath, Sampath & Millay 2018, Skeletal Muscle). Similarly, in *Drosophila* myoblast fusion there is no evidence that fusion causes transcriptional reprogramming (reviewed in Kim et al., 2015; Schejter, 2016; Aguilar et al., 2013). Last, extensive evidence in diverse tissues in *C. elegans* mutants for fusogens EFF-1 and AFF-1 demonstrate that differentiation occurs before fusion, and unfused mutant maintain differentiated state even in genetic mosaics (reviewed in Brukman et al., 2019). This kind of comparisons should be discussed in the Ms.
4. The authors used many different inhibitors (e.g. to block endocytosis, AMPK activity). However, they did not show whether these inhibitors affected VSV-G trafficking, localization, and activity. It is not clear whether the negative controls (unfused cells) express VSV-G and it appears that the authors did not check whether the inhibitors affected cell-cell fusion. Better controls for the VSV G expression and the low-pH treatment should be added: pH-insensitive VSV G mutants, temperature sensitive VSV G(ts045) mutants, fusogens that do not require low pH (e.g. Syncytins). Are the non-fused cells expressing VSV G or are they untransfected?

5. Cell cycle arrest before cell fusion in diverse organisms and in different organs has been well-characterized. However, it is also generally accepted that cells in a petri dish can fuse under the right conditions (e.g. by expressing biological or chemical fusogens) and in general these cells can divide again after they fuse. It is possible that the conditions and cells used in the current manuscript are not very permissive for division after fusion. Controls should be presented and also the effects of VSVG on cell cycle should be explored. In addition, YAP has been shown to affect organ growth, dynamics of cell growth, proliferation and cell size. More controls should be performed to determine whether the interesting observations presented by the authors are cause or effect of VSVG overexpression.

Lu, X., et al. (2018). Fine-Tuned and Cell-Cycle-Restricted Expression of Fusogenic Protein Syncytin-2 Maintains Functional Placental Syncytia. *Cell Rep* 23, 3979.

Mugahid, D., et al., (2020). YAP regulates cell size and growth dynamics via non-cell autonomous mediators. *Elife* 9.

Minor comments:

1. The transient decrease in [ATP] is significant but small. The intracellular [ATP] in cells is 5-10 mM and a small reduction in this concentration is probably not a "low energetic state". Probably a micromolar ATP concentration could cause transcriptional-reprogramming. The authors could use drugs to determine if this small reduction is sufficient to cause these changes.
2. Lines 99, 649, 1048; Figure 1; Supplementary Videos 2 and 4: Is it EGFP-GPI or mEmerald-GPI? Explain how the stable line was made or its origin.
3. Figure 1D; lines 104-106: Add a quantification of the cases where fusion occurs in the area near the coverslip.
4. Lines 109-111; Supplementary video 4 (legend): Specify in both places that U2Os cells are used in this particular experiment.
5. Lines 112-117 This has also been observed for viral-induced cell-cell fusion including VSV G - mediated cell-cell fusion. Additional references should be mentioned here.
6. Lines 124-130; For many other cell lines cell proliferation after fusion has been shown.
7. Lines 137-138; The authors should review the literature. In physiological syncytial systems (e.g. muscles in mouse and Drosophila, placenta, C. elegans' vulva and skin) the nuclei do not flow to the center of the syncytia and the transcriptional reprogramming has not been described.
8. Lines 139-165. This section is overinterpreted and should be toned down.
9. Line 172. "cyLce"
10. Lines 196-197: This is a nice result, but it is overinterpreted. It would be more convincing to show systems in which fusion is actively happening and compare fused with unfused cells (e.g. fusing myoblasts in Drosophila, zebrafish muscles, or in primary mouse myoblasts).
11. Lines 210-211: The formation of protrusions is expected to increase the surface area, not decrease it. Please discuss this.
12. Supplementary figure 1b: the color-coding is not clear. Determine a labeling for distinguishing Donor and Receiving cells.
13. Supplementary figure 4a; Lines 222-224: The internalization of the membrane dye is expected over time in normal conditions. Compare to non-fused cells.
- 14 Supplementary figure 4b: are the differences significant? Statistical analysis is required.
15. Figure 4g: Statistic analysis is missing.
16. Figure 5: Is cell division (quantified as supplementary figure 2b) still arrested when endocytosis is inhibited?

17. Line 256 (Liu et al., 2017) should be added to references.
18. Lines 263-264: the cues could be non-cell autonomous. (e.g. Mugahid, D., et al., (2020)).
19. Lines 289-291: It would be nice to follow endocytosis of VSV-G
20. Lines 293-294; Supplementary figure 5 (legend): Specify in both places that HEK cells are used in this particular experiment.
21. Figure 6a,b: Include quantification of Glut1 internalization.
22. 310-312 related to Figure 6: Endocytosis and membrane remodeling is an energy costing process by itself. Show a clear link between the glucose transporter internalization and the decrease in the cytoplasmic glucose levels. Does the Inhibition of endocytosis prevent the low energy state? Would overexpression of the Glut1 affect the proposed pathway? Is this a real low-energy state?
23. Line 355-356: References should be given to support the opening statement of the discussion. In my opinion, currently there is no in vivo evidence to support this.
24. Line 896: Explain the composition of the Fusion Buffer.
25. Line 898: State the microscope used for imaging.
26. Line 953: what does it mean "fused" and "non-fused" here? Induced and un-induced with Fusion buffer? This should also be clarified in another part of the Ms.
27. Line 1040: State the origin of the Stable line.
28. - The authors failed to reference recent literature on the mechanisms of cell fusion and the links between cell fusion and transcriptional reprogramming. The few reviews that they cite are not very recent.
29. The "hallmarks" of in vivo fusing systems have been debated in the field of cell fusion and have been mentioned several times in the manuscript. Maybe the authors should specify what are these hallmarks based on the extensive literature on cell fusion.
30. The movies are gorgeous!
31. There are many links between endocytosis and cell fusion. Maybe the authors can discuss them: Verma, S.K., Leikina, E., Melikov, K., and Chernomordik, L.V. (2014). Late stages of the synchronized macrophage fusion in osteoclast formation depend on dynamin. *Biochem J* 464, 293-300.
Shin, N.Y., Choi, H., Neff, L., Wu, Y., Saito, H., Ferguson, S.M., De Camilli, P., and Baron, R. (2014). Dynamin and endocytosis are required for the fusion of osteoclasts and myoblasts. *J Cell Biol* 207, 73-89.
Smurova, K., and Podbilewicz, B. (2016). RAB-5- and DYNAMIN-1-Mediated Endocytosis of EFF-1 Fusogen Controls Cell-Cell Fusion. *Cell reports* 14, 1517-1527.
Leikina, E., Melikov, K., Sanyal, S., Verma, S.K., Eun, B., Gebert, C., Pfeifer, K., Lizunov, V.A., Kozlov, M.M., and Chernomordik, L.V. (2013). Extracellular annexins and dynamin are important for sequential steps in myoblast fusion. *The Journal of cell biology* 200, 109-123.

Reviewer #2 (Remarks to the Author):

This manuscript describes a novel signaling program that cells initiate upon undergoing cell-cell fusion. The authors use an elegant system to precisely synchronize cell fusion, which allows them to quantitatively characterize the cellular and biochemical changes that follow. They show that cell fusion triggers reduction in cell surface area (plasma membrane) via endocytosis, which reduces the levels of plasma membrane glucose transporters resulting in reduced levels of cellular glucose and ATP. Decreased ATP levels cause an increase in cAMP dependent kinase (AMPK) activity, which decreases YAP activity (presumably by inhibiting LATS kinase). The reduction in YAP activity causes

transcriptional reprogramming and a block in cell proliferation. They demonstrate these phenomena not just in transformed cell lines but also in normal tissue that undergoes cell fusion to become multinucleate. These results imply that the diverse cell types that undergo cell fusion may utilize a common mechanism and not rely strictly on extrinsic factors. I think this work is quite novel and there would be broad interest in this manuscript.

Main Point.

The experiments are convincing and well controlled. I have a few minor concerns (see below), but I have one major issue that should be addressed. The effects of cell fusion on the signaling events upstream of YAP are relatively short lived. The changes in endocytosis, glucose, ATP, and AMPK, all normalize within an hour, yet the change in YAP regulation persists for at least 96 hours. This is not even discussed in the text. One could argue that the upstream factors “flip a switch”, which leads to persistent YAP inhibition. This possibility is testable. For example, one of the upstream events could be stimulated in normal (unfused) cells (ie. AMPK could be activated) for one hour and then the cells could be examined to see if effects on YAP persisted over a much longer term as observed after cell fusion. If long term effects on YAP were not observed, then it seems likely that other regulatory mechanisms might take over at later stages, which would be fine. I think a relatively simple experiment such as this should be done and the larger issue needs to be discussed.

Minor Points

1) In Figure 3F the staining for YAP nuclear localization in tissue sections with syncytial and normal cells is not terribly convincing. This is especially true for Part I. These experiments are also not quantified and there is not a nuclear marker, so it is hard to assess YAP nuclear localization.

2) It should be stated in the text that “YAP phosphorylation” refers to YAP phosphorylation by the LATS kinase and not AMPK.

3) Line 256: There is a reference listed (Liu et al. 2017) that is not formatted correctly in the text and not cited in the reference list.

Reviewer #3 (Remarks to the Author):

Feliciano et al. report an intriguing and detailed study of the impact of cell fusion on cellular signaling and regulation. Using a viral protein mediated fusion system they fused primarily transformed cells and observed robust signaling and structural changes. They deduced that rapid halting of cell cycle was accompanied by alteration in membrane internalization, large transcriptional changes they determined to be downstream of YAP signaling, and propose a molecular mechanism whereby glucose uptake reduction activates AMPK which impacts YAP signaling. In general, the cell biology and phenomenology of the studies is very well done, showing clearly via microscopy techniques that fusion is having a large role on membrane fusion and dynamics amongst other findings. One area where the report has less clear and robust findings is the mechanism involving glucose transport and AMPK. Some questions can be raised that if addressed would improve the conclusions around this mechanism of action:

1) The use of the glucose sensor to confirm glucose transport deficits is not ideal. As this measure is non-quantitative and is reflective of transport and metabolism of glucose, and measures levels of glucose rather than flux or transport, a more traditional measurement of 2-deoxyglucose uptake and

accumulation as 2-deoxyglucose-phosphate would be very helpful.

2) The transient effects of fusion on AMPK activity are rather modest and it is somewhat surprising that this modest change would be responsible for such large transcriptional changes. Additionally, the use of compound c, which has unclear specificity for AMPK alone, is not helpful. More relevant would be the use of AMPK null cells to confirm that AMPK is required for these changes. Additionally, it would be useful to show the impact of similar levels of AMPK activation with pharmacological activators to confirm that this degree of AMPK activation can truly elicit this broad transcriptional program.

Reviewer #4 (Remarks to the Author):

Feliciano et al present studies on cell fusion in an in vitro assay. AiryScan, TIRF and Lattice light-sheet microscopy are employed to measure changes in membrane, the cellular volume and surface area, and clathrin mediated endocytosis in cells undergoing fusion. This review will only address the use of microscope tools, not the involved cell biological aspects of this study.

Overall, I would say judicious use of advanced microscope tools was shown in this manuscript, and the different microscope systems appear adequate for this work. It is noted that in most microscope images and videos, the background appears to have been clipped aggressively. This is particularly visible in Figure 1a and the cross-sectional images in Figure 1 c-d. While it will likely not change the conclusions the authors derive from these images, I would recommend not to clip the background, as it artificially increases the signal to background of the data.

Generally, more information about the parameters for the different imaging modalities would be appreciated. As such, the video legends mostly lack acquisition times for each volume and duration of the image acquisition (Suppl. Video 4 has this information, which was appreciated). More details in the methods about the different microscope modalities are also needed. E.g. Which lattice light-sheet (square or hexagonal) was used, what were the parameters of the annulus used? I was also surprised to read that the volumetric acquisition rate was 30s per stack. Was it really that slow or is this a typo (or was a pause inserted after acquisition of a stack)?

Supplementary Video4 is of high quality and over an impressive duration for an AiryScan instrument (no notable bleaching). Was bleach correction applied in the postprocessing? If so, please indicate if this was the case for any datasets.

Despite its beauty, I fail to see the dramatic changes the authors mention that are correlated with membrane remodeling. Would this be more evident by showing the actin cytoskeleton before cell fusion?

I was surprised that in supplementary video 6, the remodeling of the plasma membrane and the corresponding measurements of volume and surface area, the AiryScan microscope was used (instead of the lattice light-sheet). I can see that the AiryScan microscope has a smaller overall PSF, leading to more precise measurements of the surface, but I would have argued that the lattice light-sheet would have faster volumetric acquisition rate, allowing finer temporal sampling (and acquisition of more timepoints). Maybe the authors can comment.

Overall, I think the use of microscopy is adequate to address the questions investigated by the

authors.

Sincerely,

RESPONSE TO REVIEWERS

Reviewer #1 (Remarks to the Author):

The work by Feliciano and co-authors addresses the question of which cellular processes are regulated as a consequence of cell-cell fusion. Using cancer cells in culture, the authors show that viral VSV G-induced cell fusion and the increased plasma membrane surface triggers a homeostatic mechanism of clathrin mediated endocytosis; a low energy cellular state which activates AMPK pathway; the re-localization of YAP1 to the cytoplasm affecting the expression of diverse genes; and a cell-cycle arrest. The study appears to be carefully conducted and the results are interesting. However, there are major related issues that the authors have to address before I can recommend it for publication.

Major comments:

1. A major concern is the generalization of the results obtained in very specific experimental conditions: cell-cell fusion induced by the viral fusogen VSV G and low pH employing mainly the breast cancer cell line, SUM-159. The authors should perform more experiments to demonstrate that the processes described in this study are conserved in other systems (e.g. different cell types, including stem and primary cells lines) and cell fusion events induced by different ways (e.g. fusogens that do not require acidification, chemical/electrical fusion). Alternatively, the authors could re-write the manuscript to avoid making general conclusions. Even though this occurs throughout the Ms. the generalization is reflected clearly in the title, which should have some additions: "Cell fusion (induced by VSV G) triggers membrane diminution causing transcriptional reprogramming (in cancer cells in vitro)".

We have addressed this concern in two ways. While the editor indicated that we would not be required to examine additional cell types, we wanted to verify that the key observation, that YAP1 is retained in the cytoplasm after cell fusion, is also true in a different cell fusion model. We performed fusion experiments using the C2C12 myoblast cell line that fuses to form myotubes in culture. The results, now presented in Figure 3, show that in dividing C2C12 cells YAP1 is present in the nucleus. In contrast, in C2C12 fused myofibers, YAP1 is inhibited and largely excluded from the nucleus. This is consistent with our results using VSV-G mediated fusion of SUM-159 as well as our findings in both primary human trophoblasts fusion and myoblast fusion during mouse muscle development and supports the hypothesis that the change in YAP1 localization occurs not only after VSV-G mediated cell fusion, but also following syncytial formation induced by other fusogens.

While these additional results give us confidence that cell fusion in many contexts results in nuclear exclusion of YAP1 we recognize that our mechanistic dissection has been performed only using VSV-G-induced fusion. As suggested by the reviewer, we have changed the title and have modified many conclusions to more clearly communicate which findings we cannot yet generalize.

2. The studies of VSV G as a fusogen are classic and researchers are still actively working to understand the mechanisms and consequences of VSV G-mediated fusion. The authors have not cited this extensive work in their introduction and also not in the discussion (for example White et al., 1981; Abou-Hamdan et al., 2018)

We appreciate the reviewer highlighting this oversight. We have added 3 new references on this topic. We hope these will help interested readers find more information about the foundational studies investigating the mechanisms of VSV-G-mediated fusion.

3. The present work is based on a gain-of-function system to show a link between a cell-cell fusion event and the up-regulation of genes related to differentiation (down-regulation of genes related to proliferation). However, some loss-of-function studies reported that knockout myoblasts for Myomaker or Myomerger fail to fuse but still differentiate, suggesting that fusion does not directly regulate differentiation (reviewed in Sampath, Sampath & Millay 2018, Skeletal Muscle). Similarly, in *Drosophila* myoblast fusion there is no evidence that fusion causes transcriptional reprogramming (reviewed in Kim et al., 2015; Schejter, 2016; Aguilar et al., 2013). Last, extensive evidence in diverse tissues in *C. elegans* mutants for fusogens EFF-1 and AFF-1 demonstrate that differentiation occurs before fusion, and unfused mutant maintain differentiated state even in genetic mosaics (reviewed in Brukman et al., 2019). This kind of comparisons should be discussed in the Ms.

The reviewer brings up *in vivo* investigations into several important fusion systems and they allow us to add some interesting comparisons to the discussion. In tissue systems, fusion happens in the context of complex extracellular signaling environments. The presence of some differentiation features in Myomaker and Myomerger knockout cells is likely due, at least in part, to extracellular cues. Similarly, in *C. elegans*, differentiation that occurs prior to fusion is likely due to non-cell autonomous signaling. As the reviewer points out, the strategy used in this study is a gain-of-function system. It allows us to interrogate consequences of cell-cell fusion in the absence of extracellular signals.

The insights from the study presented in this manuscript might inspire future experiments that directly compare the complete transcriptional signatures of fusion-deficient mutants with the wt-fused population in living systems to define the contributions of extracellular signaling programs to differentiation. Recent studies that have defined distinct transcriptional signatures of fused and unfused cell populations in both skeletal muscle and placenta (Petranj M.J. *et al.*, 2020, Rubenstein A. B. *et al.*, 2020, Liu Y. *et al.*, 2018) could be used as important references for such investigations.

4. The authors used many different inhibitors (e.g. to block endocytosis, AMPK activity). However, they did not show whether these inhibitors affected VSV-G trafficking, localization, and activity. It is not clear whether the negative controls (unfused cells) express VSV-G and it appears that the authors did not check whether the inhibitors affected cell-cell fusion. Better controls for the VSV G expression and the low-pH

treatment should be added: pH-insensitive VSV G mutants, temperature sensitive VSV G(ts045) mutants, fusogens that do not require low pH (e.g. Syncytins). Are the non-fused cells expressing VSV G or are they untransfected?

The reviewer raises several important points about the controls used in the study and we are pleased to expand and clarify this dimension of our investigation.

To address whether inhibitors used in this study (PitStop2, Dynasore, and compound C) alter VSV-G localization and function we performed new experiments. We transfected SUM-159 cells with VSV-G-EGFP and compared the cellular distribution VSV-G-EGFP in inhibitor-treated and untreated cells. The results presented in Supplementary Figure 8 show that VSV-G-EGFP cellular localization is similar in all conditions, which indicates that the overall trafficking of VSV-G is not affected. We also note in the text that these results are “consistent with the observed ability of inhibitor treated cells to fuse.”

An additional issue raised by the reviewer is whether the non-fused cells used as negative controls express VSV-G. The short answer is that the majority are. The transfection efficiency of VSV-G in our experiments is about 60% (we added this detail to the Method section). A feature of the VSV-G fusion system is that because only one side of the fusion pair requires VSV-G for fusion, untransfected cells can participate in fusion events. We improved the description of the system in the first paragraph of the result section.

We also performed experiments to control for the effect of VSV-G and low-pH and found no effect on the proliferation state of cells. These are presented in supplementary figure 3 and described in more detail in response to the 5th comment from this reviewer.

We recognize that some of the reviewer’s concerns may have been driven by confusing nomenclature. Previously we used the term “non-fused” to describe different conditions in different contexts. In the revised manuscript, control cells transfected with VSV-G but analyzed without an acid wash to induce fusion are referred to as “control”. The term “non-fused” refers to VSV-G transfected cells in the same dish as fused cells. These cells were washed with fusion buffer and did not fuse (either because they were distant from other cells or perhaps due to transfection inefficiency). Control cells not transfected with VSV-G are specified as “- VSV-G”.

5. Cell cycle arrest before cell fusion in diverse organisms and in different organs has been well-characterized. However, it is also generally accepted that cells in a petri dish can fuse under the right conditions (e.g. by expressing biological or chemical fusogens) and in general these cells can divide again after they fuse. It is possible that the conditions and cells used in the current manuscript are not very permissive for division after fusion. Controls should be presented and also the effects of VSVG on cell cycle should be explored. In addition, YAP has been shown to affect organ growth, dynamics of cell growth, proliferation and cell size. More controls should be performed to determine whether the interesting observations presented by the authors are cause or

effect of VSVG overexpression.

Before we jump into a discussion of the experiments performed to address the reviewer's concerns, we want to clarify how we understand the terms division and proliferation (or cell cycle) in the context of syncytial systems. In the process of syncytial division, nuclei are distributed as the plasma membrane segments portions of the shared cytoplasm generating new, smaller cells. In contrast, proliferation could also be described as nuclear division and it involves cell cycle hallmarks such as DNA duplication, chromosome alignment and separation of genetic material into daughter nuclei. Such nuclear cycling has been investigated in multinucleate systems such as the developing *Drosophila* embryo. The presence of P21 indicates nuclei are not able to proliferate.

To address the reviewer's question whether SUM-159 cells are competent to divide after VSV-G-mediated fusion, we fused cells and monitored syncytia for 24 hours. In addition to SUM-159 cells, identical experiments were also performed using U2OS and 293T cells. Very few syncytia divided and in all three cell types >85% of syncytia remained intact. These results (presented in supplementary Figure 4) confirm that the cells used in these experiments are minimally permissive for syncytial division after fusion.

To address the reviewers concerns that VSV-G expression could alter proliferation in SUM-159 cells we tested whether VSV-G expression in the absence of fusion buffer or the fusion buffer treatment of untransfected cells altered the fraction of cells cycling. The results of these experiments (presented in supplementary figure 3) indicate that neither VSV-G expression nor the fusion buffer can account for the observed increase in P21 positive nuclei shown in Figure 1f.

Lu, X., et al. (2018). Fine-Tuned and Cell-Cycle-Restricted Expression of Fusogenic Protein Syncytin-2 Maintains Functional Placental Syncytia. *Cell Rep* 23, 3979.
Mugahid, D., et al., (2020). YAP regulates cell size and growth dynamics via non-cell autonomous mediators. *Elife* 9.

Minor comments:

1. The transient decrease in [ATP] is significant but small. The intracellular [ATP] in cells is 5-10 mM and a small reduction in this concentration is probably not a "low energetic state". Probably a micromolar ATP concentration could cause transcriptional-reprogramming. The authors could use drugs to determine if this small reduction is sufficient to cause these changes.

As an energy sensor AMPK is activated in response to reductions in energy state (levels of AMP, ADP and ATP). Our results showing that activation of AMPK correlates with a decrease in ATP levels upon cell fusion, strongly suggests a reduced energy state in fused cells. As suggested, we used drugs to activate AMPK and assayed whether this was sufficient to shift the distribution of YAP1. These experiments are also

related to suggestions from Reviewers 2 and 3 and the new results are shown in Figure 7g, h. Activating AMPK was sufficient to cause a change in the distribution of YAP1, however the phenotype wasn't as robust as the change observed when cells fused. We now discuss that activation of AMPK due to a reduced energy state is necessary but not sufficient, and cell fusion is required to cause the long-term changes.

2. Lines 99, 649, 1048; Figure 1; Supplementary Videos 2 and 4: Is it EGFP-GPI or mEsmerald-GPI? Explain how the stable line was made or its origin.

mEsmerald-GPI was transfected together with VSV-G. We appreciate the opportunity to correct this mistake. These experiments were not performed using a stable cell line but using transient transfection.

3. Figure 1D; lines 104-106: Add a quantification of the cases where fusion occurs in the area near the coverslip.

Fusion pore formation near the coverslip was observed in all cells imaged. We have stated this in the results section.

4. Lines 109-111; Supplementary video 4 (legend): Specify in both places that U2Os cells are used in this particular experiment.

As suggested by the reviewer we have clarified this in the manuscript.

5. Lines 112-117 This has also been observed for viral-induced cell-cell fusion including VSV G -mediated cell-cell fusion. Additional references should be mentioned here.

Nuclear clustering not only occurs in physiological syncytia systems, but also occurs in other *in vitro* models. We agreed with the reviewer and have expanded the references on this topic

6. Lines 124-130; For many other cell lines cell proliferation after fusion has been shown.

As stated by the reviewer several studies of *in vitro* cell fusion have shown that in some circumstance cells can divide after they have fused. However, most of these cells died after they attempt to divide and only 1% can continue dividing (Duelli, D. & Lazebnik, Y., 2003). As we stated in major comment # 5, we have performed additional experiments using U2OS and HEK 293T cells and found that, similar to SUM-159 cells, about 90% of the syncytia formed after VSV-G fusion did not divide. These findings parallel previous studies and are in agreement with the idea that after cell fusion most syncytia cannot divide. The results section now includes this information.

7. Lines 137-138; The authors should review the literature. In physiological syncytial systems (e.g. muscles in mouse and Drosophila, placenta, C. elegans' vulva and skin)

the nuclei do not flow to the center of the syncytia and the transcriptional reprogramming has not been described.

We appreciate the reviewer comments on this topic. We have included several references showing the formation of nuclear clusters during cell fusion in skeletal muscle, placenta, and macrophage fusion. In the case of cell fusion in muscle, there is extensive literature characterizing the regulation of nuclei positioning during muscle development, a process regulated by the Linker of Nucleoskeleton and Cytoskeleton (LINC) complex (Cadot, B. *et al*, 2015). We have also added new references showing evidence of transcriptional reprogramming after cell fusion.

8. Lines 139-165. This section is overinterpreted and should be toned down.

We have modified the text in this section to focus on describing the results reported.

9. Line 172. “cyLCe”

We have corrected this in the manuscript.

10. Lines 196-197: This is a nice result, but it is overinterpreted. It would be more convincing to show systems in which fusion is actively happening and compare fused with unfused cells (e.g. fusing myoblasts in *Drosophila*, zebrafish muscles, or in primary mouse myoblasts).

We agree with the reviewer that live imaging of different physiological cell fusion models would contribute greatly to our findings. However, catching these fusion events *in vivo* is extremely difficult, and would require new transgenic models expressing YAP1 tagged with a fluorescent protein.

11. Lines 210-211: The formation of protrusions is expected to increase the surface area, not decrease it. Please discuss this.

We have revised this part in the manuscript to specify that the protrusions observed were formed immediately after cells fused, but they disappear thereafter. This decrease in membrane protrusions was accompanied by an overall decrease in surface area, which we have shown is mediated by an increase in endocytosis.

12. Supplementary figure 1b: the color-coding is not clear. Determine a labeling for distinguishing Donor and Receiving cells.

We have added in the legend a note to specify that curves in black and orange, showing decrease in fluorescence, are from Donor cells.

13. Supplementary figure 4a; Lines 222-224: The internalization of the membrane dye is

expected over time in normal conditions. Compare to non-fused cells.

As suggested by the reviewer, we added images and the quantitation of the number of vesicles internalized in control non-fusing cells treated with Fusion Buffer to the supplementary figure (now Supplementary Figure 7a).

14 Supplementary figure 4b: are the differences significant? Statistical analysis is required.

These differences are significant and we have included the statistical analysis. This is now Supplementary figure 7b.

15. Figure 4g: Statistic analysis is missing.

We have added the statistical analysis.

16. Figure 5: Is cell division (quantified as supplementary figure 2b) still arrested when endocytosis is inhibited?

This is an interesting question. We performed this experiment and, surprisingly, we observed that after inhibition of endocytosis with PitStop syncytial division, but not proliferation (nuclear cycling), increases. However, we have not included these results as part of this manuscript. We may include it in future studies.

17. Line 256 (Liu et al., 2017) should be added to references.

We have corrected this in the manuscript.

18. Lines 263-264: the cues could be non-cell autonomous. (e.g. Mugahid, D., et al., (2020)).

We agree with the reviewer that YAP1 activity can be regulated by non-cell autonomous cues. It is additionally conceivably that there are cues released into the media during fusion (a creative and interesting idea). However, because this sentence is in the context of endocytosis, it is appropriate to describe endocytosis as a cell-intrinsic cue.

19. Lines 289-291: It would be nice to follow endocytosis of VSV-G

This is an interesting topic, but unfortunately was outside of the focus of the manuscript. Although, we thank the reviewer for the suggestion.

20. Lines 293-294; Supplementary figure 5 (legend): Specify in both places that HEK cells are used in this particular experiment.

As suggested by the reviewer we have clarified the cell type. In the revised manuscript, this is now in Supplementary Figure 9b.

21. Figure 6a,b: Include quantification of Glut1 internalization.

We have included the quantification of Glut1 internalization as part of Supplementary Figure 9.

22. 310-312 related to Figure 6: Endocytosis and membrane remodeling is an energy costing process by itself. Show a clear link between the glucose transporter internalization and the decrease in the cytoplasmic glucose levels. Does the Inhibition of endocytosis prevent the low energy state? Would overexpression of the Glut1 affect the proposed pathway? Is this a real low-energy state?

The reviewer correctly identifies energy consuming processes that occur after cell fusion. We hypothesize that the internalization of Glut1 slows the ability of the cell to generate additional ATP in response to these expenditures. If the increase in expenditure was the only cause of the observed decrease in ATP levels, we would not expect to find the correlating decreases in cytoplasmic glucose that are quantified in figure 6. We conclude that inhibition of endocytosis prevents the low energy state because treatment with Pitstop prevents the rapid increase in phosphorylated AMPK (shown in Figure 7b).

The comment raised here is also related to questions we address in response to Reviewer 3.

23. Line 355-356: References should be given to support the opening statement of the discussion. In my opinion, currently there is no in vivo evidence to support this.

As suggested by the reviewer, we have included references supporting this opening statement.

24. Line 896: Explain the composition of the Fusion Buffer.

As suggested by the reviewer, we have included the composition of the Fusion Buffer in the Methods section.

25. Line 898: State the microscope used for imaging.

We have taken care to be sure that the microscopes used in each figure and video are clearly communicated in the legends.

26. Line 953: what does it mean “fused” and “non-fused” here? Induced and un-induced with Fusion buffer? This should also be clarified in another part of the Ms.

We have specified in the manuscript that “Fused” refers to cells expressing VSV-G that have been washed with Fusion buffer to induced cell fusion and “non-fused” cells are cells in the same dish that did not fuse as described in response to Comment #4 above.

27. Line 1040: State the origin of the Stable line.

We have included a note in the method section that describes how these cells were generated in our lab.

28. - The authors failed to reference recent literature on the mechanisms of cell fusion and the links between cell fusion and transcriptional reprogramming. The few reviews that they cite are not very recent.

We have revised the manuscript and incorporated additional references including recent work describing transcriptional reprogramming after cell fusion in both muscle and placenta.

29. The “hallmarks” of in vivo fusing systems have been debated in the field of cell fusion and have been mentioned several times in the manuscript. Maybe the authors should specify what are these hallmarks based on the extensive literature on cell fusion.

We now specify in the last paragraph of the introduction that the traditional “hallmarks” we assay in this study are fusion pore formation, plasma membrane mixing and cytoplasmic mixing. As a result of these investigations we propose that depletion of YAP1 and downstream differentiation be considered as additional, emerging hallmarks.

30. The movies are gorgeous!

We appreciate the reviewer comment.

31. There are many links between endocytosis and cell fusion. Maybe the authors can discuss them:

This is an interesting topic. We edited the discussion section to include this topic.

Verma, S.K., Leikina, E., Melikov, K., and Chernomordik, L.V. (2014). Late stages of the synchronized macrophage fusion in osteoclast formation depend on dynamin. *Biochem J* 464, 293-300. Shin, N.Y., Choi, H., Neff, L., Wu, Y., Saito, H., Ferguson, S.M., De Camilli, P., and Baron, R. (2014). Dynamin and endocytosis are required for the fusion of osteoclasts and myoblasts. *J Cell Biol* 207, 73-89. Smurova, K., and Podbilewicz, B. (2016). RAB-5- and DYNAMIN-1-Mediated Endocytosis of EFF-1 Fusogen Controls Cell-Cell Fusion. *Cell reports* 14, 1517-1527. Leikina, E., Melikov, K., Sanyal, S., Verma, S.K., Eun, B., Gebert, C., Pfeifer, K., Lizunov, V.A., Kozlov, M.M., and Chernomordik, L.V. (2013). Extracellular annexins and dynamin are important for sequential steps in myoblast fusion. *The Journal of cell biology* 200, 109-123.

Reviewer #2 (Remarks to the Author):

This manuscript describes a novel signaling program that cells initiate upon undergoing cell-cell fusion. The authors use an elegant system to precisely synchronize cell fusion, which allows them to quantitatively characterize the cellular and biochemical changes that follow. They show that cell fusion triggers reduction in cell surface area (plasma membrane) via endocytosis, which reduces the levels of plasma membrane glucose transporters resulting in reduced levels of cellular glucose and ATP. Decreased ATP levels cause an increase in cAMP dependent kinase (AMPK) activity, which decreases YAP activity (presumably by inhibiting LATS kinase). The reduction in YAP activity causes transcriptional reprogramming and a block in cell proliferation. They demonstrate these phenomena not just in transformed cell lines but also in normal tissue that undergoes cell fusion to become multinucleate. These results imply that the diverse cell types that undergo cell fusion may utilize a common mechanism and not rely strictly on extrinsic factors. I think this work is quite novel and there would be broad interest in this manuscript.

Main Point.

The experiments are convincing and well controlled. I have a few minor concerns (see below), but I have one major issue that should be addressed. The effects of cell fusion on the signaling events upstream of YAP are relatively short lived. The changes in endocytosis, glucose, ATP, and AMPK, all normalize within an hour, yet the change in YAP regulation persists for at least 96 hours. This is not even discussed in the text. One could argue that the upstream factors “flip a switch”, which leads to persistent YAP inhibition. This possibility is testable. For example, one of the upstream events could be stimulated in normal (unfused) cells (ie. AMPK could be activated) for one hour and then the cells could be examined to see if effects on YAP persisted over a much longer term as observed after cell fusion. If long term effects on YAP were not observed, then it seems likely that other regulatory mechanisms might take over at later stages, which would be fine. I think a relatively simple experiment such as this should be done and the larger issue needs to be discussed.

We thank the reviewer for this excellent suggestion. To address it we used two compounds to activate AMPK (AICAR or 2-DG for 1hr) and assessed the degree of YAP1 cytoplasmic redistribution in unfused cells. Our results in Figure 7g and 7h show that activation of AMPK, alone, in unfused cells does shift the nuclear to cytoplasmic ratio of YAP1 but does not elicit the same robust YAP1 cytoplasmic localization phenotype observed after cell fusion. As the reviewer indicated, these results suggest that cell fusion triggers additional regulatory mechanisms that, together with AMPK activation, cause persistent nuclear exclusion of YAP1.

To further address the larger issue, we now discuss that AMPK contributes to flipping a switch. We highlight previous studies showing that similar treatment of unfused cells with AICAR and 2-DG can induce partial cytoplasmic localization of YAP1 that was reversed after drug removal (DeRan et al., 2014; Mo et al., 2015; Wang et al., 2015). These changes enhance the paper and reveal that AMPK activation helps initiate

redistribution of YAP1 away from the nucleus and that the long-lasting change that facilitates transcriptional reprogramming upon fusion is likely promoted by additional regulatory mechanisms.

Minor Points

1) In Figure 3F the staining for YAP nuclear localization in tissue sections with syncytial and normal cells is not terribly convincing. This is especially true for Part I. These experiments are also not quantified and there is not a nuclear marker, so it is hard to assess YAP nuclear localization.

We thank the reviewers for pointing this out. We agree that it is difficult to discriminate YAP1 localization, especially in the fused cells. We revised the mouse muscle images and moved them to supplemental Fig. 6. It is clear that YAP1 is present in the nucleus of the progenitor cells (PAX7). In fused cells (MF20), YAP1 staining is very low. This could be because inactive YAP1 in the cytoplasm has been degraded (Wu, Z. & Guan K., 2021).

We replaced the tissue images in figure 3 with new experiments examining the localization of YAP1 in cultured muscle cells, C2C12 cells, that fuse upon treatment with differentiation media. These cells more clearly demonstrate that the changes in YAP1 inhibition and localization observed upon VSV-G-induced fusion can also occur in cells that fuse using endogenous fusogens.

2) It should be stated in the text that “YAP phosphorylation” refers to YAP phosphorylation by the LATS kinase and not AMPK.

As requested by the reviewer we have clarified in the manuscript that “phosphorylation of YAP1” refers to the YAP1 phosphorylation by the LATS kinase at S127.

3) Line 256: There is a reference listed (Liu et al. 2017) that is not formatted correctly in the text and not cited in the reference list.

We have corrected this in the manuscript.

Reviewer #3 (Remarks to the Author):

Feliciano et al. report an intriguing and detailed study of the impact of cell fusion on cellular signaling and regulation. Using a viral protein mediated fusion system they fused primarily transformed cells and observed robust signaling and structural changes. The

deduced that rapid halting of cell cycle was accompanied by alteration in membrane internalization, large transcriptional changes they determined to be downstream of YAP signaling, and propose a molecular mechanism whereby glucose uptake reduction activates AMPK which impacts YAP signaling. In general, the cell biology and phenomenology of the studies is very well done, showing clearly via microscopy techniques that fusion is having a large role on membrane fusion and dynamics amongst other findings. One area where the report has less clear and robust findings is the mechanism involving glucose transport and AMPK. Some questions can be raised that if addressed would improve the conclusions around this mechanism of action:

We appreciate the reviewer's compliments about the study, and we are pleased to have the opportunity to enhance and clarify the investigations into mechanisms that lead to changes in YAP1 signaling and transcription. Through our efforts to address the questions of all of the reviewers, we improved our understanding of the contribution of glucose transport and AMPK signaling upon cell fusion. We expand on this below, but briefly, the revised manuscript communicates that the energy sensing pathway is likely supported by additional regulatory mechanisms that help sustain changes in YAP1 localization.

1) The use of the glucose sensor to confirm glucose transport deficits is not ideal. As this measure is non-quantitative and is reflective of transport and metabolism of glucose, and measures levels of glucose rather than flux or transport, a more traditional measurement of 2-deoxyglucose uptake and accumulation as 2-deoxyglucose-phosphate would be very helpful.

We thought this was an excellent recommendation, so we performed experiments using a fluorescently labeled 2-deoxy-glucose analog (2-NBDG) to measure glucose transport in fused and non-fused cells. We realized, however, that this experiment is more complicated due to the dynamic nature of fusing cells. Ideally, we would want to measure any changes in 2-NBDG uptake as transporters are internalized and returned to the surface following fusion. However, because the glucose analog is not hydrolysable, the fluorescence assay measures the accumulation of 2-NBDG. Assessing changes in transport rates were further complicated because 2-NBDG was also taken up through endocytosis. We modified the text to communicate that the limitation of the glucose sensor we utilized is that it only reports the cytoplasmic glucose levels and not flux.

2) The transient effects of fusion on AMPK activity are rather modest and it is somewhat surprising that this modest change would be responsible for such large transcriptional changes. Additionally, the use of compound c, which has unclear specificity for AMPK alone, is not helpful. More relevant would be the use of AMPK null cells to confirm that AMPK is required for these changes. Additionally, it would be useful to show the impact of similar levels of AMPK activation with pharmacological activators to confirm that this degree of AMPK activation can truly elicit this broad transcriptional program.

As we understand it, the reviewer recommends we clarify whether AMPK is necessary and sufficient for transcriptional outcome. These valuable comments are related to those raised by other reviewers as well.

To address the issue of necessity we did additional experiments, so we now employ two strategies to disable AMPK activity during fusion: Compound C and the dominant negative form of the AMPK $\alpha 2$ subunit. As the reviewer indicates, long term incubation with compound C has been shown to have off target effects. We were aware of this limitation and reduced the incubation time of compound C to 3 hrs in order to minimize any off-target effects. We also agreed that a second line of experimentation would be valuable. We assayed YAP1 distribution before and after fusion of cells transfected with a dominant negative form of the $\alpha 2$ subunit of AMPK (AMPK-DK). While the expression of AMPK-DK largely inhibited YAP1, blocking its nuclear exclusion, it is possible that some fraction of the endogenous $\alpha 1$ AMPK subunits are still able to partially promote YAP1 inhibition. Overall, this approach was more expedient than generating the suggested AMPK null cells and had the advantage of being a more acute manipulation. This new data was included in figure 7c, d.

To address the issue of sufficiency we assayed the effect of AMPK activation (using AICAR or 2DG) on YAP1 localization in the absence of fusion. This data is now included in the revised Figure 7g, h. As detailed in comments to other reviewers and in the discussion, these results demonstrate that AMPK activation alone is not sufficient to cause persistent exclusion of YAP1 from the nucleus.

Based on the new experiments we conclude that the change in AMPK is necessary, but not sufficient to cause the long-lasting changes. We suggest that the rapid - and admittedly modest - change in AMPK may be physiologically relevant based on similarity to other systems where similar rapid and modest changes in AMPK activity is known to influence cellular programs (including neuronal tolerance of glutamate excitation, myocardial ischemia, and cardiomyocyte growth (1.5-4 fold activation in 5-10 min) (Weisova, P. *et al.*, 2009, Takano, A.P. *et al.*, 2013)).

Reviewer #4 (Remarks to the Author):

Feliciano et al present studies on cell fusion in an in vitro assay. AiryScan, TIRF and Lattice light-sheet microscopy are employed to measure changes in membrane, the cellular volume and surface area, and clathrin mediated endocytosis in cells undergoing fusion. This review will only address the use of microscope tools, not the involved cell biological aspects of this study.

Overall, I would say judicious use of advanced microscope tools was shown in this manuscript, and the different microscope systems appear adequate for this work. It is noted that in most microscope images and videos, the background appears to have been

clipped aggressively. This is particularly visible in Figure 1a and the cross-sectional images in Figure 1 c-d. While it will likely not change the conclusions the authors derive from these images, I would recommend not to clip the background, as it artificially increases the signal to background of the data.

We thank the reviewer for this comment. In all figures the intention of the Hoechst staining was to identify the location of the nucleus. We processed images using macros that include steps to eliminate unwanted structures labeled by Hoechst staining. These structures were mostly debris from dead cells in the cultures. We have included a note for this step in the Methods section. The background was not clipped in any other channels.

Generally, more information about the parameters for the different imaging modalities would be appreciated. As such, the video legends mostly lack acquisition times for each volume and duration of the image acquisition (Suppl. Video 4 has this information, which was appreciated).

As requested, we have included information about the imaging parameters to the Supplementary video legends.

More details in the methods about the different microscope modalities are also needed. E.g. Which lattice light-sheet (square or hexagonal) was used, what were the parameters of the annulus used?

We have included the additional details including the requested specifications in the Methods section.

I was also surprised to read that the volumetric acquisition rate was 30s per stack. Was it really that slow or is this a typo (or was a pause inserted after acquisition of a stack)?

The exposure time was 0.018s and there were 250 slices = 4.8s. As mentioned by the reviewer we included a pause between stacks to acquire one stack every 30s to minimize phototoxicity/photobleaching. This has been included in the Methods section.

Supplementary Video4 is of high quality and over an impressive duration for an AiryScan instrument (no notable bleaching). Was bleach correction applied in the postprocessing? If so, please indicate if this was the case for any datasets.

Thank you. Even though photobleaching was minimal, we applied a histogram matching photobleaching correction. We have included this information in the method section as requested.

Despite its beauty, I fail to see the dramatic changes the authors mention that are correlated with membrane remodeling. Would this be more evident by showing the actin cytoskeleton before cell fusion?

Supplementary Figure 2 now includes Pre-Fusion (before fusion), Fusion (during) and Post-Fusion (after fusion) still images from Supplementary Video 4 highlighting changes in the actin cytoskeleton induced by cell fusion.

I was surprised that in supplementary video 6, the remodeling of the plasma membrane and the corresponding measurements of volume and surface area, the AiryScan microscope was used (instead of the lattice light-sheet). I can see that the AiryScan microscope has a smaller overall PSF, leading to more precise measurements of the surface, but I would have argued that the lattice light-sheet would have faster volumetric acquisition rate, allowing finer temporal sampling (and acquisition of more timepoints). Maybe the authors can comment.

The assessment of changes in surface area and volume upon fusion were pioneering experiments as we began to investigate the VSV-G-mediated cell fusion system. While we agree that lattice light sheet microscopy would have advantages, we felt that the measurements obtained from LSM 880 imaging demonstrate robust differences sufficient for publication.

Overall, I think the use of microscopy is adequate to address the questions investigated by the authors.

Sincerely,
Reto Fiolka

REVIEWERS' COMMENTS

Reviewer #1 (Remarks to the Author):

In this revised manuscript, Feliciano et al. have addressed the comments of the reviewers and the result is an improvement in the message, addition of quantitative analyses of fusion and the authors also discuss the literature in a more balanced way. However, there are still major issues that the authors have to address before I can recommend it for publication.

Major comments:

1. A major conclusion of the paper is that fusion triggers transcriptional reprogramming of cells, however, this is not supported by the current cell-cell fusion literature. The hypothesis that fusion reprogrammed cells assumes that if fusion fails then the cells will not differentiate and their fates will be defective. However, many mutations that result in fusion failure, including mutations in the fusion machinery (fusogens) of the placenta, muscles, macrophages and epithelia in mice, zebrafish, *Drosophila*, *Arabidopsis* and *C. elegans* do not affect the cell fate of the somatic unfused cells [1-21].
2. The evidence from many in vivo and in vitro systems is that cells exit the cell cycle before they fuse and that these programmed cells differentiate before they express the fusogens on their surface. Therefore, the hypothesis that cell fusion reprograms cells needs stronger evidence than what the authors provide. In addition, even cells in culture receive extrinsic signals from other cells and from the serum therefore the conclusion that “cell fusion promotes transcriptional-reprogramming independent from extrinsic cues” is flawed.
3. When VSVG induces fusion in cultured cells it is a pathological event and these fusing cells are probably in different stages in the cell cycle. It has been shown that fusion of cells in different stages in the cell cycle respond differently. In contrast, cells in the same stage in the cell cycle (e.g. G0) will not have to reprogram themselves. The YAP1 nuclear efflux could be related to cell cycle conflicts between the fusing cells and not to syncytial reprogramming. This may also be a problem in the interpretation of the RNAseq data. See for example this classic studies: [22, 23]. Moreover, fusion of proliferating cells with B cells result in hybridomas that instead of exiting the cell cycle they become proliferative. Thus, since fusion failure does not result in proliferation of unfused cells and in vitro, cells exit the cell cycle (G0) before they fuse, then cell fusion does not lead to cell cycle exit.

Lines 50-51 The cells exit cell cycle before they fuse

Lines 54-57 There are no references and this is hypothetical

Lines 63-65 This is not true, developmental geneticists test cell autonomous versus non-cell autonomous phenomena in vivo using genetic mosaics, cell-specific degradation, cell-sp RNAseq, in situ hybridization, IF, cell specific rescue and many other approaches.

Lines 66-67 As mentioned above, serum and neighboring cells provide many cues in cell cultures.

Lines 70-72 This is not something new, it has been shown before for VSVG and for many other fusogens.

Lines 74-75 This is an overinterpretation

Line 77 The acute reduction in ATP is probably a very mild reduction of ~15%

Lines 132-134 They lose the competence to enter the cell cycle before they fuse!

Lines 147-152 Many fused cells divide and fusion in cultured cells result in major chromosomal aberrations including aneuploidy, polyploidy and cell cycle conflicts that can result in cancer [24-30].

Lines 191-228. A study from last year showed that YAP negatively regulates cell fusion in the

placenta and represses genes that promote trophoblast fusion [10]. Feliciano et al. may be analyzing a defense mechanism against polyploidy and/or pathologically induced cell-cell fusion that induces the reduction in YAP activity. Then YAP directly inhibits genes that promote cell fusion (as shown in the placenta).

The changes in membrane dynamics following fusion have been reviewed and discussed in the literature of cell-cell fusion [31-38].

Experiments using RNAi, knockout and overexpression of wild-type and mutant YAP1 are required to strengthen the model proposed in this manuscript and to determine whether YAP1 dynamics is cause or effect. Is the proposed transcriptional reprogramming triggered by cell fusion?

Many references support that in vivo cells get their fate before they fuse and failure to fuse does not change the fate of the cells. The references in Line 415 are not convincing and do not support the claim.

Lines 429-435 Citations to papers from Drosophila and zebrafish are missing and the results are overinterpreted

Lines 436-447 This paragraph is misleading, and the literature is misrepresented. Single cell RNAseq, multiple transcriptomics and numerous studies in mice, zebrafish, Drosophila, zebrafish, C. elegans and tissue culture cells have identified many genes that specify the differentiated cell stages before fusion. Many transcription factors have intrinsic activities that control expression of cell-cell fusogens (see reviews: [39-44])

Figures 1, 3, 4, 5, 7, S3, S5, S6, S9:

How many independent experiments?

N=?

What statistical tests were performed? The readers will not check the excel tables to figure this out...The legends should include this information

Figure S4 Polyploidy, aneuploidy, nuclear fusion will block cell cycle

A section on statistics is missing in materials and methods

Minor comments:

1. Line 38 assemble "lipidic" fusion pores
2. Line 39 SARS-CoV2 instead of COVID-19
3. Line 40 eukaryotic
4. Line 92-93 A reference is missing.
5. Lines 93-96 This has been shown before for many cell-cell fusion events. The characterization performed by the authors is very well performed but it is not unique and references should be added here.
6. Lines 106-107 This is also something that has been observed before for many cell-cell fusion reactions
7. Lines 123-124 many authors have shown actin remodeling during cell-cell fusion (references are missing, for example: [2, 33, 45-49])
8. Lines 137-139 Confluence and contact inhibition may be involved here.
9. Lines 340-341 References should be provided
10. Lines 405-408 The results suggest. This is an overinterpretation of the results
11. Lines 450-452 The conclusion is different from what the authors say. In Ref #74 it was shown that continuous and constitutive internalization of the EFF-1 fusogen before and after fusion represses fusion. Only transient surface expression in both fusing cells mediates fusion.
12. Lines 484-488. Discuss [10] here.
13. Lines 491-493 This is an overinterpretation. It is necessary to show this in KO animals to show

that YAP1 inhibition is essential for a syncytial state triggered by fusion.

14. Line 948 immunostained with

15. Line 982 differences

16. Lines 1140-1142: How many independent experiments were performed

17. Figure 1 Kinetics of cell-cell fusion should be compared to other systems. For example: [2, 21, 31, 50-52]

18. Figure 3 Is the data from a single experiment? How many times was this experiment performed? In panels d and f the significance should be above the data points.

19. Figure 5 Difficult to see the difference between Hoechst and P21 in panel C. Are the nuclei in different stages of the cell cycle? The experiment with synchronized cells or using FACS could determine if the cells are in different stages of the cell cycle

20. Figure 6e the differences in ATP levels are very small. A standard curve could help determining the ATP concentration inside the cells. Negative controls for times 5, 15 and 60 min?

21. Figure 7g Are all the nuclei in the same stage of the cell cycle (G0, G1, S?)

ATP depletion with KCN and Azide could help.

References

1. Rushton E, Drysdale R, Abmayr SM, Michelson AM, Bate M. Mutations in a novel gene, myoblast city, provide evidence in support of the founder cell hypothesis for *Drosophila* muscle development. *Development*. 1995;121:1979-88.
2. Richardson BE, Beckett K, Nowak SJ, Baylies MK. SCAR/WAVE and Arp2/3 are crucial for cytoskeletal remodeling at the site of myoblast fusion. *Development*. 2007;134(24):4357-67. PubMed PMID: 18003739.
3. Aguilar PS, Baylies MK, Fleissner A, Helming L, Inoue N, Podbilewicz B, et al. Genetic basis of cell-cell fusion mechanisms. *Trends in genetics : TIG*. 2013;29:427-37. doi: 10.1016/j.tig.2013.01.011. PubMed PMID: 23453622.
4. Schafer G, Weber S, Holz A, Bogdan S, Schumacher S, Muller A, et al. The Wiskott-Aldrich syndrome protein (WASP) is essential for myoblast fusion in *Drosophila*. *Dev Biol*. 2007;304(2):664-74. Epub 2007/02/20. doi: S0012-1606(07)00031-0 [pii] 10.1016/j.ydbio.2007.01.015. PubMed PMID: 17306790.
5. Dottermusch-Heidel C, Groth V, Beck L, Onel SF. The Arf-GEF Schizo/Loner regulates N-cadherin to induce fusion competence of *Drosophila* myoblasts. *Dev Biol*. 2012;368(1):18-27. doi: 10.1016/j.ydbio.2012.04.031. PubMed PMID: 22595515.
6. Chen EH, Pryce BA, Tzeng JA, Gonzalez GA, Olson EN. Control of myoblast fusion by a guanine nucleotide exchange factor, loner, and its effector ARF6. *Cell*. 2003;114(6):751-62. PubMed PMID: 14505574.
7. Chen EH, Olson EN. Antisocial, an intracellular adaptor protein, is required for myoblast fusion in *Drosophila*. *Developmental Cell*. 2001;1:705-15.
8. Dhanyasi N, Segal D, Shimoni E, Shinder V, Shilo BZ, VijayRaghavan K, et al. Surface apposition and multiple cell contacts promote myoblast fusion in *Drosophila* flight muscles. *J Cell Biol*. 2015;211(1):191-203. doi: 10.1083/jcb.201503005. PubMed PMID: 26459604; PubMed Central PMCID: PMC4602036.
9. Massarwa R, Carmon S, Shilo BZ, Schejter ED. WIP/WASp-based actin-polymerization machinery is essential for myoblast fusion in *Drosophila*. *Dev Cell*. 2007;12(4):557-69. PubMed PMID: 17419994.
10. Meinhardt G, Haider S, Kunihs V, Saleh L, Pollheimer J, Fiala C, et al. Pivotal role of the transcriptional co-activator YAP in trophoblast stemness of the developing human placenta. *Proceedings of the National Academy of Sciences of the United States of America*. 2020;117(24):13562-70. PubMed PMID: Medline:32482863.

11. Mi S, Lee X, Li X-p, Veldman GM, Finnerty H, Racie L, et al. Syncytin is a captive retroviral envelope protein involved in human placental morphogenesis. *Nature*. 2000;403:785-9.
12. Dupressoir A, Vernochet C, Bawa O, Harper F, Pierron G, Opolon P, et al. Syncytin-A knockout mice demonstrate the critical role in placentation of a fusogenic, endogenous retrovirus-derived, envelope gene. *Proceedings of the National Academy of Sciences of the United States of America*. 2009;106(29):12127-32. Epub 2009/07/01. doi: 10.1073/pnas.0902925106. PubMed PMID: 19564597; PubMed Central PMCID: PMC2715540.
13. Maruyama D, Volz R, Takeuchi H, Mori T, Igawa T, Kurihara D, et al. Rapid Elimination of the Persistent Synergid through a Cell Fusion Mechanism. *Cell*. 2015;161(4):907-18. doi: 10.1016/j.cell.2015.03.018. PubMed PMID: 25913191.
14. Motomura K, Kawashima T, Berger F, Kinoshita T, Higashiyama T, Maruyama D. A pharmacological study of Arabidopsis cell fusion between the persistent synergid and endosperm. *J Cell Sci*. 2018;131(2). doi: 10.1242/jcs.204123. PubMed PMID: 28808086.
15. Mohler WA, Shemer G, del Campo J, Valansi C, Opoku-Serebuoh E, Scranton V, et al. The type I membrane protein EFF-1 is essential for developmental cell fusion in *C. elegans*. *Dev Cell*. 2002;2(3):355-62. PubMed PMID: 11879640.
16. Shemer G, Suissa M, Kolotuev I, Nguyen KCQ, Hall DH, Podbilewicz B. EFF-1 is sufficient to initiate and execute tissue-specific cell fusion in *C. elegans*. *Curr Biol*. 2004;14(17):1587-91. PubMed PMID: 15341747.
17. Sapir A, Choi J, Leikina E, Avinoam O, Valansi C, Chernomordik LV, et al. AFF-1, a FOS-1-Regulated Fusogen, Mediates Fusion of the Anchor Cell in *C. elegans*. *Dev Cell*. 2007;12(5):683-98. PubMed PMID: 17488621.
18. Zhang W, Roy S. Myomaker is required for the fusion of fast-twitch myocytes in the zebrafish embryo. *Dev Biol*. 2017;423(1):24-33. doi: 10.1016/j.ydbio.2017.01.019. PubMed PMID: 28161523.
19. Millay DP, O'Rourke JR, Sutherland LB, Bezprozvannaya S, Shelton JM, Bassel-Duby R, et al. Myomaker is a membrane activator of myoblast fusion and muscle formation. *Nature*. 2013;499(7458):301-5. doi: 10.1038/nature12343. PubMed PMID: 23868259; PubMed Central PMCID: PMC3739301.
20. Leikina E, Gamage DG, Prasad V, Goykhberg J, Crowe M, Diao J, et al. Myomaker and Myomerger Work Independently to Control Distinct Steps of Membrane Remodeling during Myoblast Fusion. *Dev Cell*. 2018;46(6):767-80 e7. doi: 10.1016/j.devcel.2018.08.006. PubMed PMID: 30197239; PubMed Central PMCID: PMC6203449.
21. Gattegno T, Mittal A, Valansi C, Nguyen KC, Hall DH, Chernomordik LV, et al. Genetic control of fusion pore expansion in the epidermis of *Caenorhabditis elegans*. *Mol Biol Cell*. 2007;18(4):1153-66. PubMed PMID: 17229888.
22. Rao PN, Johnson RT. Mammalian cell fusion: studies on the regulation of DNA synthesis and mitosis. *Nature*. 1970;225:159-64.
23. Johnson RT, Rao PN. Mammalian cell fusion: induction of premature chromosome condensation in interphase nuclei. *Nature*. 1970;226(5247):717-22. PubMed PMID: 5443247.
24. Larsson LI, Bjerregaard B, Wulf-Andersen L, Talts JF. Syncytin and cancer cell fusions. *TheScientificWorldJournal*. 2007;7:1193-7. doi: 10.1100/tsw.2007.212. PubMed PMID: 17704852.
25. Duelli DM, Padilla-Nash HM, Berman D, Murphy KM, Ried T, Lazebnik Y. A virus causes cancer by inducing massive chromosomal instability through cell fusion. *Curr Biol*. 2007;17(5):431-7. PubMed PMID: 17320392.
26. Duelli D, Lazebnik Y. Cell fusion: A hidden enemy? *Cancer Cell*. 2003;3(5):445-8. PubMed PMID: 12781362.
27. Uygur B, Leikina E, Melikov K, Villasmil R, Verma SK, Vary CPH, et al. Interactions with Muscle Cells Boost Fusion, Stemness, and Drug Resistance of Prostate Cancer Cells. *Mol Cancer Res*.

2019;17(3):806-20. doi: 10.1158/1541-7786.MCR-18-0500. PubMed PMID: 30587522.

28. Lazova R, Laberge GS, Duvall E, Spoelstra N, Klump V, Sznol M, et al. A Melanoma Brain Metastasis with a Donor-Patient Hybrid Genome following Bone Marrow Transplantation: First Evidence for Fusion in Human Cancer. *PLoS One*. 2013;8(6):e66731. doi: 10.1371/journal.pone.0066731. PubMed PMID: 23840523; PubMed Central PMCID: PMC3694119.

29. Pawelek JM, Chakraborty AK. Fusion of tumour cells with bone marrow-derived cells: a unifying explanation for metastasis. *Nat Rev Cancer*. 2008;8(5):377-86. Epub 2008/04/04. doi: nrc2371 [pii] 10.1038/nrc2371. PubMed PMID: 18385683.

30. Chakraborty A, Lazova R, Davies S, Backvall H, Ponten F, Brash D, et al. Donor DNA in a renal cell carcinoma metastasis from a bone marrow transplant recipient. *Bone Marrow Transplant*. 2004;34(2):183-6. Epub 2004/06/15. doi: 10.1038/sj.bmt.1704547 1704547 [pii]. PubMed PMID: 15195072.

31. Mohler WA, Simske JS, Williams-Masson EM, Hardin JD, White JG. Dynamics and ultrastructure of developmental cell fusions in the *Caenorhabditis elegans* hypodermis. *Curr Biol*. 1998;8:1087-90. PubMed PMID: 9768364.

32. Linton C, Neumann B, Giordano-Santini R, Hilliard MA. RAB-5 regulates regenerative axonal fusion by controlling EFF-1 endocytosis. *bioRxiv*2018.

33. Haralalka S, Shelton C, Cartwright HN, Guo F, Trimble R, Kumar RP, et al. Live imaging provides new insights on dynamic F-actin filopodia and differential endocytosis during myoblast fusion in *Drosophila*. *PLoS One*. 2014;9(12):e114126. doi: 10.1371/journal.pone.0114126. PubMed PMID: 25474591; PubMed Central PMCID: PMC4256407.

34. Soulavie F, Hall DH, Sundaram MV. The AFF-1 exoplasmic fusogen is required for endocytic scission and seamless tube elongation. *Nat Commun*. 2018;9(1):1741. doi: 10.1038/s41467-018-04091-1. PubMed PMID: 29717108; PubMed Central PMCID: PMC5931541.

35. Soulavie F, Sundaram MV. Auto-fusion and the shaping of neurons and tubes. *Seminars in cell & developmental biology*. 2016;60:136-45. doi: 10.1016/j.semcd.2016.07.018. PubMed PMID: 27436685.

36. Shemer G, Podbilewicz B. Fusomorphogenesis: Cell fusion in organ formation. *Dev Dyn*. 2000;218:30-51. PubMed PMID: 10822258.

37. Shin NY, Choi H, Neff L, Wu Y, Saito H, Ferguson SM, et al. Dynamin and endocytosis are required for the fusion of osteoclasts and myoblasts. *J Cell Biol*. 2014;207(1):73-89. doi: 10.1083/jcb.201401137. PubMed PMID: 25287300; PubMed Central PMCID: PMC4195819.

38. Sundaram MV, Cohen JD. Time to make the doughnuts: Building and shaping seamless tubes. *Seminars in cell & developmental biology*. 2017;67:123-31. doi: 10.1016/j.semcd.2016.05.006. PubMed PMID: 27178486; PubMed Central PMCID: PMC45104681.

39. Martin SG. Role and organization of the actin cytoskeleton during cell-cell fusion. *Seminars in cell & developmental biology*. 2016;60:121-6. doi: 10.1016/j.semcd.2016.07.025. PubMed PMID: 27476112.

40. Giordano-Santini R, Linton C, Hilliard MA. Cell-cell fusion in the nervous system: Alternative mechanisms of development, injury, and repair. *Seminars in cell & developmental biology*. 2016;60:146-54. doi: 10.1016/j.semcd.2016.06.019. PubMed PMID: 27375226.

41. Iosilevskii Y, Podbilewicz B. Programmed cell fusion in development and homeostasis. *Curr Top Dev Biol*. 2021;144. doi: <https://doi.org/10.1016/bs.ctdb.2020.12.013>.

42. Goh Q, Millay DP. Requirement of myomaker-mediated stem cell fusion for skeletal muscle hypertrophy. *eLife*. 2017;6. doi: 10.7554/eLife.20007. PubMed PMID: 28186492.

43. Shinn-Thomas JH, Mohler WA. New insights into the mechanisms and roles of cell-cell fusion. *Int Rev Cell Mol Biol*. 2011;289:149-209. Epub 2011/07/14. doi: 10.1016/B978-0-12-386039-2.00005-5. PubMed PMID: 21749901.

44. Chen EH, Grote E, Mohler W, Vignery A. Cell-cell fusion. *FEBS Lett.* 2007;581:2181-93. PubMed PMID: 17395182.
45. Chen A, Leikina E, Melikov K, Podbilewicz B, Kozlov MM, Chernomordik LV. Fusion-pore expansion during syncytium formation is restricted by an actin network. *J Cell Sci.* 2008;121(Pt 21):3619-28. PubMed PMID: 18946025.
46. Kim S, Shilagardi K, Zhang S, Hong SN, Sens KL, Bo J, et al. A critical function for the actin cytoskeleton in targeted exocytosis of prefusion vesicles during myoblast fusion. *Dev Cell.* 2007;12(4):571-86. PubMed PMID: 17419995.
47. Zhang Y, Yang Y, Zhu Z, Ou G. WASP-Arp2/3-dependent actin polymerization influences fusogen localization during cell-cell fusion in *Caenorhabditiselegans* embryos. *Biology open.* 2017;6(9):1324-8. doi: 10.1242/bio.026807. PubMed PMID: 28760733; PubMed Central PMCID: PMC5612239.
48. Gruenbaum-Cohen Y, Harel I, Umansky KB, Tzahor E, Snapper SB, Shilo BZ, et al. The actin regulator N-WASp is required for muscle-cell fusion in mice. *Proceedings of the National Academy of Sciences of the United States of America.* 2012;109(28):11211-6. Epub 2012/06/28. doi: 10.1073/pnas.1116065109. PubMed PMID: 22736793; PubMed Central PMCID: PMC3396508.
49. Mukherjee P, Gildor B, Shilo BZ, VijayRaghavan K, Schejter ED. The actin nucleator WASp is required for myoblast fusion during adult *Drosophila* myogenesis. *Development.* 2011;138(11):2347-57. Epub 2011/05/12. doi: 10.1242/dev.055012. PubMed PMID: 21558381; PubMed Central PMCID: PMC3091497.
50. Richard JP, Leikina E, Langen R, Henne WM, Popova M, Balla T, et al. Intracellular curvature-generating proteins in cell-to-cell fusion. *Biochem J.* 2011;440(2):185-93. doi: 10.1042/BJ20111243. PubMed PMID: 21895608; PubMed Central PMCID: PMC3216009.
51. Hu C, Ahmed M, Melia TJ, Sollner TH, Mayer T, Rothman JE. Fusion of cells by flipped SNAREs. *Science.* 2003;300(5626):1745-9. PubMed PMID: 12805548.
52. Reeves JD, Gallo SA, Ahmad N, Miamidian JL, Harvey PE, Sharron M, et al. Sensitivity of HIV-1 to entry inhibitors correlates with envelope/coreceptor affinity, receptor density, and fusion kinetics. *Proc Natl Acad Sci U S A.* 2002;99(25):16249-54. doi: 10.1073/pnas.252469399. PubMed PMID: 12444251; PubMed Central PMCID: PMC138597.

Reviewer #2 (Remarks to the Author):

The authors have sufficiently addressed my primary concern, and I think the manuscript is suitable for publication.

Reviewer #3 (Remarks to the Author):

The authors have addressed many concerns and improved the quality of the conclusions they are able to draw with this work.

Reviewer #1 (Remarks to the Author):

In this revised manuscript, Feliciano et al. have addressed the comments of the reviewers and the result is an improvement in the message, addition of quantitative analyses of fusion and the authors also discuss the literature in a more balanced way. However, there are still major issues that the authors have to address before I can recommend it for publication.

Major comments:

1. A major conclusion of the paper is that fusion triggers transcriptional reprogramming of cells, however, this is not supported by the current cell-cell fusion literature. The hypothesis that fusion reprogrammed cells assumes that if fusion fails then the cells will not differentiate and their fates will be defective. However, many mutations that result in fusion failure, including mutations in the fusion machinery (fusogens) of the placenta, muscles, macrophages and epithelia in mice, zebrafish, *Drosophila*, *Arabidopsis* and *C. elegans* do not affect the cell fate of the somatic unfused cells [1-21].

The revised manuscript now highlights that transcriptional reprogramming follows cell fusion. To address the reviewer's comment, we first want to clarify that in our view, functional differentiation of syncytia *in vivo* not only requires the unification of cells through cell fusion, but it is also driven by external tissue-specific cues. This is in agreement with recent transcriptomic studies in muscle and placenta demonstrating that fusion-competent cells and their corresponding syncytia have different transcriptional signatures. As the reviewer has pointed out, several studies have shown that fusion-defective cells can acquire some features of differentiated cells. We disagree that these findings demonstrate cell fusion is a nonessential step for reprogramming. Rather, any differentiation that occurs in the absence of fusion is attributable we believe to other signals. Furthermore, studies in mouse and human cells have demonstrated that impaired cell fusion leads to defects in the development and regeneration of muscle and affect placenta function. This demonstrates that even though fusion-defective cells can partially acquire characteristics of differentiated cells, they are not functionally equivalent to their syncytial counterpart.

Our transcriptomics analysis revealed that after inducing cell fusion the transcription of several genes involved in cell proliferation dropped, and genes involved in cell-cycle arrest (i.e., P21, *CDKN1A*) and differentiation were expressed. These findings are in agreement with the role of cell fusion in syncytia reprogramming. We have nevertheless toned down our emphasis on a direct effect of cell fusion on physiological reprogramming of syncytia, acknowledging our mechanistic dissection has been performed only using VSV-G-induced fusion.

2. The evidence from many *in vivo* and *in vitro* systems is that cells exit the cell cycle before they fuse and that these programmed cells differentiate before they express the fusogens on their surface. Therefore, the hypothesis that cell fusion reprograms cells needs stronger evidence than what the authors provide. In addition, even cells in culture receive extrinsic signals from other cells and from the serum therefore the conclusion

that “cell fusion promotes transcriptional-reprogramming independent from extrinsic cues” is flawed.

As stated by the reviewer, cells in culture receive extrinsic signals from other cells and from the serum in culture media. Since unfused and fused cells are cultured in the same media in our VSV-G fusion experiments, it is unlikely that cues from the serum are responsible for driving reprogramming. Furthermore, our findings show that only cells induced to fuse by low pH undergo transcriptional changes, demonstrating that these changes are driven by the act of cell fusion.

It is true that studies in muscle and placenta have suggested fusion-competent cells express P21 and exit the cell-cycle in order to commit to fusion. However, it is possible these cells re-enter the cell cycle when the necessary cues for differentiation are not provided and growth signals are present. Recent studies have shown that in many cell types P21, at moderate levels, induce cell-cycle arrest. When the levels of P21 are not further increased, terminal differentiation signals are not deployed and cells can re-enter the cell-cycle (Zhao *et al.* Cell Reports, 2020). Interestingly, P21 in muscle and placenta cell systems reaches its highest levels after formation of syncytia compared to intermediate levels in fusion-competent cells. It is possible that additional intrinsic cues provided by the merging of cells could further increase P21 levels thus promoting terminal differentiation programs. Our results showing increases in P21 levels after structural changes coincident with cell fusion are in agreement with this idea.

3. When VSVG induces fusion in cultured cells it is a pathological event and these fusing cells are probably in different stages in the cell cycle. It has been shown that fusion of cells in different stages in the cell cycle respond differently. In contrast, cells in the same stage in the cell cycle (e.g. G0) will not have to reprogram themselves. The YAP1 nuclear efflux could be related to cell cycle conflicts between the fusing cells and not to syncytial reprogramming. This may also be a problem in the interpretation of the RNAseq data. See for example this classic studies: [22, 23]. Moreover, fusion of proliferating cells with B cells result in hybridomas that instead of exiting the cell cycle they become proliferative. Thus, since fusion failure does not result in proliferation of unfused cells and in vitro, cells exit the cell cycle (G0) before they fuse, then cell fusion does not lead to cell cycle exit.

Lines 50-51 The cells exit cell cycle before they fuse

Lines 54-57 There are no references and this is hypothetical

Lines 63-65 This is not true, developmental geneticists test cell autonomous versus non-cell autonomous phenomena in vivo using genetic mosaics, cell-specific degradation, cell-sp RNAseq, in situ hybridization, IF, cell specific rescue and many other approaches.

Lines 66-67 As mentioned above, serum and neighboring cells provide many cues in cell cultures.

Lines 70-72 This is not something new, it has been shown before for VSVG and for many other fusogens.

Lines 74-75 This is an overinterpretation

Line 77 The acute reduction in ATP is probably a very mild reduction of ~15%

Lines 132-134 They lose the competence to enter the cell cycle before they fuse!
Lines 147-152 Many fused cells divide and fusion in cultured cells result in major chromosomal aberrations including aneuploidy, polyploidy and cell cycle conflicts that can result in cancer [24-30].

Lines 191-228. A study from last year showed that YAP negatively regulates cell fusion in the placenta and represses genes that promote trophoblast fusion [10]. Feliciano et al. may be analyzing a defense mechanism against polyploidy and/or pathologically induced cell-cell fusion that induces the reduction in YAP activity. Then YAP directly inhibits genes that promote cell fusion (as shown in the placenta).

The changes in membrane dynamics following fusion have been reviewed and discussed in the literature of cell-cell fusion [31-38].

Experiments using RNAi, knockout and overexpression of wild-type and mutant YAP1 are required to strengthen the model proposed in this manuscript and to determine whether YAP1 dynamics is cause or effect. Is the proposed transcriptional reprogramming triggered by cell fusion?

Many references support that in vivo cells get their fate before they fuse and failure to fuse does not change the fate of the cells. The references in Line 415 are not convincing and do not support the claim.

Lines 429-435 Citations to papers from Drosophila and zebrafish are missing and the results are overinterpreted

Lines 436-447 This paragraph is misleading, and the literature is misrepresented. Single cell RNAseq, multiple transcriptomics and numerous studies in mice, zebrafish, Drosophila, zebrafish, C. elegans and tissue culture cells have identified many genes that specify the differentiated cell stages before fusion. Many transcription factors have intrinsic activities that control expression of cell-cell fusogens (see reviews: [39-44]

Figures 1, 3, 4, 5, 7, S3, S5, S6, S9:

How many independent experiments?

N=?

What statistical tests were performed? The readers will not check the excel tables to figure this out...The legends should include this information

Figure S4 Polyploidy, aneuploidy, nuclear fusion will block cell cycle

A section on statistics is missing in materials and methods

The reviewer raises several important points, which we have summarized into four categories: 1) The cell cycle state of cells before fusion, 2) studies of cell intrinsic cues within organisms, 3) clarity of findings and statistics, and 4) citations. Other topics are discussed in Minor comments to avoid redundancies.

- 1) As the reviewer points out, cells in culture are not perfectly synchronized with respect to their cell-cycle state. Based on our results in Figure 1h, we estimate a small fraction of about 15% of cells are in M phase at a given time. Because fusion of cells in conflicting phases of the cell-cycle (i.e., fusion of cells in M phase with cells in interphase) are known to die, they should lose adhesion and be largely washed out and excluded from our analysis. We have included a note to clarify this in the Results and Methods sections.

- 2) Despite the effort for dissecting cell autonomous vs non-cell autonomous phenomena *in vivo* using genetic mosaics and other approaches, there are many limitations due to the challenge of isolating the intrinsic contribution of cell fusion from ever-present signaling molecules in organisms. Here, we have attempted to circumvent these constraints by employing a VSV-G fusogen-mediated assay to induce cell fusion of culture cells in the absence of tissue-specific cues. This has allowed us to interrogate the consequences of cell fusion in the absence of these signals. On the other hand, new insights on the direct role of non-autonomous signals for syncytia differentiation can be interrogated *in vivo* using fusion-defective systems in combination with transcriptomics. We have discussed this in the manuscript.
- 3) As mentioned in comment #1, impaired cell fusion leads to defects in muscle and placenta function demonstrating cell fusion is indispensable for the formation of functional syncytial systems. In addition, recent RNA-seq studies in human placenta and mouse skeletal muscle cells have revealed distinct transcriptional signatures between syncytia and mononucleated fusion-competent cells (Petrany M.J. *et al.*, 2020, Rubenstein A. B. *et al.*, 2020, Liu Y. *et al.*, 2018). This suggests that the final transcriptional signatures of syncytia are acquired after cell fusion. Despite this evidence, as mention in comment #1, in the revised manuscript we have toned down the idea of a direct effect of cell fusion on physiological reprogramming of syncytia, acknowledging our mechanistic dissection has been performed only using VSV-G-induced fusion. Furthermore, as suggested by the reviewer, we have included additional details of our statistical analysis in the Methods section and figures legends.
- 4) We agree that many studies have been fundamental to understand how the different steps involved in cell fusion promote the generation of functional syncytia. Therefore, we have added 21 new references including many topics suggested by the reviewer. We hope this will help interested readers find more information about different aspects of cell fusion.

Minor comments:

1. Line 38 assemble “lipidic” fusion pores

Despite cell fusion leading to the merging of plasma membranes and involving the mixing of lipids, we have not used the term “lipidic” fusion pores for clarity as this term has been extensively used in studies describing specific fusion events between vesicles and the plasma membrane.

2. Line 39 SARS-CoV2 instead of COVID-19

We have corrected this in the manuscript.

3. Line 40 eukaryotic

We have corrected this in the manuscript.

4. Line 92-93 A reference is missing.

As suggested by the reviewer, we have included references supporting this statement.

5. Lines 93-96 This has been shown before for many cell-cell fusion events. The characterization performed by the authors is very well performed but it is not unique and references should be added here.

We have made the text clear that these characteristics have been previously described in different cell fusion systems. We began our investigation with a comprehensive morphological characterization of cell fusion as this was essential to determine the timeline of changes happening in our study. This allowed us to define the role of these changes in cell fate determination of syncytia.

6. Lines 106-107 This is also something that has been observed before for many cell-cell fusion reactions

As mentioned in minor comment 5, the goal of our study was to assess all of these characteristics using the same model system for fusion (i.e., VSV-G) so we could better define their mechanistic roles in syncytia reprogramming.

7. Lines 123-124 many authors have shown actin remodeling during cell-cell fusion (references are missing, for example: [2, 33, 45-49])

As suggested by the reviewer, we have included additional references here.

8. Lines 137-139 Confluence and contact inhibition may be involved here.

The reviewer brings up an interesting point. We agree that confluence and contact inhibition could affect the extent of cell fusion in our assay. Because of this all experiments were performed at similar cell confluency. In addition, our controls experiments demonstrated that changes in P21 nuclear levels were driven by cell fusion (See Supplementary Figure 3).

9. Lines 340-341 References should be provided

As suggested by the reviewer, we have included additional references here.

10. Lines 405-408 The results suggest. This is an overinterpretation of the results

We have modified this in the revised manuscript.

11. Lines 450-452 The conclusion is different from what the authors say. In Ref #74 it was shown that continuous and constitutive internalization of the EFF-1 fusogen before and after fusion represses fusion. Only transient surface expression in both fusing cells mediates fusion.

We have rephrased this section to directly reflect the focus of the study.

12. Lines 484-488. Discuss [10] here.

As suggested by the reviewer, we have discussed this study in both Result and Discussion sections.

13. Lines 491-493 This is an overinterpretation. It is necessary to show this in KO animals to show that YAP1 inhibition is essential for a syncytial state triggered by fusion.

Due to the important roles of YAP1 during development, the use of KO YAP1 animals (even if inducible) for this purpose might lead to mix phenotypes that could be difficult to dissect.

14. Line 948 immunostained with

We have corrected this in the Supporting Information.

15. Line 982 differences

We have corrected this in the Supporting Information.

16. Lines 1140-1142: How many independent experiments were performed

We now have included this information in figure legends.

17. Figure 1 Kinetics of cell-cell fusion should be compared to other systems. For example: [2, 21, 31, 50-52]

This is an interesting topic, but unfortunately is outside of the focus of the manuscript.

18. Figure 3 Is the data from a single experiment? How many times was this experiment performed? In panels d and f the significance should be above the data points.

We now have included this information in figure legends.

19. Figure 5 Difficult to see the difference between Hoechst and P21 in panel C. Are the nuclei in different stages of the cell cycle? The experiment with synchronized cells or using FACS could determine if the cells are in different stages of the cell cycle

The panel illustrates the differences in P21 levels between Non-fused (Hoechst can be visualized) and Fused (P21 is mostly visualized) cells and compares these differences with cells treated with an endocytic inhibitor.

20. Figure 6e the differences in ATP levels are very small. A standard curve could help determining the ATP concentration inside the cells. Negative controls for times 5, 15 and 60 min?

We have only measured the relative changes in ATP levels during cell fusion and not the exact concentration, we have specified this in the manuscript.

21. Figure 7g Are all the nuclei in the same stage of the cell cycle (G0, G1, S?) ATP depletion with KCN and Azide could help.

This is an interesting topic, but unfortunately was outside of the scope of the manuscript.

Reviewer #2 (Remarks to the Author):

The authors have sufficiently addressed my primary concern, and I think the manuscript is suitable for publication.

Reviewer #3 (Remarks to the Author):

The authors have addressed many concerns and improved the quality of the conclusions they are able to draw with this work.